# Neuroimaging and behavioral evidence that violent video games exert no negative effect on human empathy for pain and emotional reactivity to violence

Lukas Leopold Lengersdorff[1], Isabella C Wagner[1], Gloria Mittmann[1], David Sastre-Yagüe[1], Andre Lüttig[1], Andreas Olsson[2], Pedrag Petrovic[2], Claus Lamm[1]*

[1]Social, Cognitive and Affective Neuroscience Unit, Department of Cognition, Emotion, and Methods in Psychology, Faculty of Psychology, University of Vienna, Vienna, Austria; [2]Department of Clinical Neuroscience, Division of Psychology, Karolinska Institute, Stockholm, Sweden

**Abstract** Influential accounts claim that violent video games (VVGs) decrease players' emotional empathy by desensitizing them to both virtual and real-life violence. However, scientific evidence for this claim is inconclusive and controversially debated. To assess the causal effect of VVGs on the behavioral and neural correlates of empathy and emotional reactivity to violence, we conducted a prospective experimental study using functional magnetic resonance imaging (fMRI). We recruited 89 male participants without prior VVG experience. Over the course of two weeks, participants played either a highly violent video game or a non-violent version of the same game. Before and after this period, participants completed an fMRI experiment with paradigms measuring their empathy for pain and emotional reactivity to violent images. Applying a Bayesian analysis approach throughout enabled us to find substantial evidence for the absence of an effect of VVGs on the behavioral and neural correlates of empathy. Moreover, participants in the VVG group were not desensitized to images of real-world violence. These results imply that short and controlled exposure to VVGs does not numb empathy nor the responses to real-world violence. We discuss the implications of our findings regarding the potential and limitations of experimental research on the causal effects of VVGs. While VVGs might not have a discernible effect on the investigated subpopulation within our carefully controlled experimental setting, our results cannot preclude that effects could be found in settings with higher ecological validity, in vulnerable subpopulations, or after more extensive VVG play.

*For correspondence:
claus.lamm@univie.ac.at

Competing interest: The authors declare that no competing interests exist.

## Editor's evaluation

Lengersdorff and colleagues present behavioural and fMRI data that are valuable in demonstrating no impact of violent video games on the emotional response to pain in their particular sample. The effects may be specific to the participant group who have no neurological disorder and no character traits that would predispose to desensitisation (because they are selected due to little prior experience playing these games), and there are some openly-discussed test-retest reliability issues (session 1->2) with the fMRI measures, but they present convincing evidence for the absence of effect in this group.

**eLife digest** Violent video games have often been accused of facilitating aggressive behaviour, in particular due to concerns that they could numb players toward real violence and therefore result in decreased empathy towards the pain of others. However, studies investigating these claims have often produced conflicting results, potentially due to methodological issues. For instance, work showing that violent games lead to emotional desensitization has often relied on testing participants immediately after a gaming session, which limits interpretations about prolonged impact. Many studies also compare gamers to people with no gaming experience, making it difficult to assess whether violent games decrease empathy, or whether less empathetic individuals are more likely to be drawn to this content.

Lengersdorff et al. aimed to examine the long-term effects of violent video games using an experimental design that would bypass some of these limitations. A group of 89 young men with little gaming experience were recruited to play either a highly or non-violent version of the same game for seven hour-long sessions over two weeks. The way their brain reacted to violent images and processed other people's pain was assessed before and after this 'gaming training' using fMRI. The analyses showed no changes in these measures in volunteers who played the violent version of the game, suggesting that it had not numbed them to violence or affected their empathy.

While experimental studies cannot fully capture the experiences of real-world gamers, the findings by Lengersdorff et al. represent a step towards resolving the scientific controversy surrounding the effects of violent games. Ultimately, a deeper understanding of how this type of media influences our emotions could help inform policymaking decisions about access to violent content.

## Introduction

Video games have evolved into one of the most popular forms of entertainment. In Europe, 25% of the population report playing video games weekly, and especially young adults spend much time in these 'virtual worlds' (*IPSOS MediaCT, 2012*). Many popular games contain high levels of violent imagery, with the killing or hurting of other characters being deeply engrained in the gameplay (*Gentile et al., 2004*; *Krantz et al., 2017*). Many recent studies have investigated whether such violent video games (VVGs) have adverse effects on real-world social behavior and empathy (*Anderson et al., 2010*). According to the influential general aggression model (*Bushman and Anderson, 2002*), VVGs should decrease the players' empathy for the pain of others by desensitizing them to both virtual and real violence. Such desensitizing effects should in turn be reflected by decreased activity in brain areas underpinning empathy, such as the anterior insula (AI) and the anterior midcingulate cortex (aMCC) (*Lamm et al., 2011*; *Lamm et al., 2019*). However, the evidence for this prediction is mixed. While some studies found that playing VVGs leads to emotional desensitization on the behavioral and neural level (*Arriaga et al., 2011*; *Bartholow et al., 2006*; *Carnagey et al., 2007*; *Engelhardt et al., 2011*; *Staude-Müller et al., 2008*), other studies failed to reveal such effects (*Gao et al., 2017*; *Kühn et al., 2018*; *Szycik et al., 2017a*; *Szycik et al., 2017b*). Conflicting results are also found on the level of systematic reviews (*de Vrieze, 2018*; *Mathur and VanderWeele, 2019*). Several meta-analyses suggest that VVGs exert small, yet consistent adverse effects on aggression and empathy (*Anderson et al., 2010*; *Calvert et al., 2017*; *Greitemeyer and Mügge, 2014*; *Mathur and VanderWeele, 2019*; *Prescott et al., 2018*). Other researchers contest these results, claiming that results are a product of selective reporting and biased analyses (*Ferguson and Kilburn, 2010*; *Hilgard et al., 2017b*).

A key question is whether VVGs are causally responsible for low empathy, or whether less empathic individuals are more likely to play VVGs (*Bushman and Anderson, 2015*; *Ferguson et al., 2008*). Many studies have been quasi-experimental in nature, comparing the empathic responses of participants who habitually play VVGs with those of participants without VVG experience (*Bartholow et al., 2005*; *Bartholow et al., 2006*; *Gentile et al., 2016*; *Krahé et al., 2011*). Such designs provide limited information on the direction of the causal link between VVGs and decreased empathy. The existing experimental studies have nearly always used VVGs as an experimental manipulation shortly before measuring the outcomes of interest (*Arriaga et al., 2011*; *Bushman and Anderson, 2009*; *Carnagey et al., 2007*; *Engelhardt et al., 2011*; *Guo et al., 2013*; *Staude-Müller et al., 2008*). While these studies consistently report evidence for a desensitizing effect of violent games, they cannot

disentangle the immediate effects of VVG play from those that have a persistent, long-term impact on individuals. Immediate VVG effects may encompass a wide range of processes, such as priming (*Bushman, 1998*), as well as stress-like responses such as increases in active fear and aggressive behaviors (*Fanselow, 1994*; *Mobbs et al., 2007*; *Mobbs et al., 2009*) that include generally increased sympathetic activity, release of stress hormones, heightened activation of involved brain structures, and cognitive-affective responses (e.g. deep reflection on the seen content, and changes in emotions and mood). Such responses can persist on a timescale of minutes to hours after aversive events such as VVG exposure, and have been shown to negatively affect social behavior (*Nitschke et al., 2022*). It is important to distinguish these immediate effects from longer-term adaptations that occur over days or weeks, such as habituation or memory consolidation processes. The general aggression model predicts that the repeated exposure to violence in the positive emotional context of videogames leads to the gradual extinction of aversive reactions, resulting in the long-term desensitization of players to real-world violence (*Bushman and Anderson, 2009*).

It is therefore essential to conduct experimental studies that can disentangle the long- and short-term effects of VVGs in participants without prior VVG experience. One first such study was conducted by *Kühn et al., 2018*, who found no significant effects of VVGs on empathy and its neural correlates. While this study was an important starting point, four important design features limited its conclusions. First, the researchers used very dissimilar games in the experimental group versus the control group, restricting the comparability of the two conditions. Second, while the participants of the experimental group were asked to play the violent game *Grand Theft Auto V* (Rockstar Studios) for 30 min per day over 2 months, the authors did not control the degree to which participants actually played the game. Third, the authors did not control that participants actually committed violent acts within the game, as the game offers a large amount of gameplay without violent content. Fourth, the absence of significant results was interpreted as evidence for the absence of VVG effects. However, the authors did not report the results of equivalence tests (*Lakens et al., 2018*) or Bayesian hypothesis tests (*Keysers et al., 2020*) that would support such claims conclusively (*Hilgard et al., 2017a*). In view of

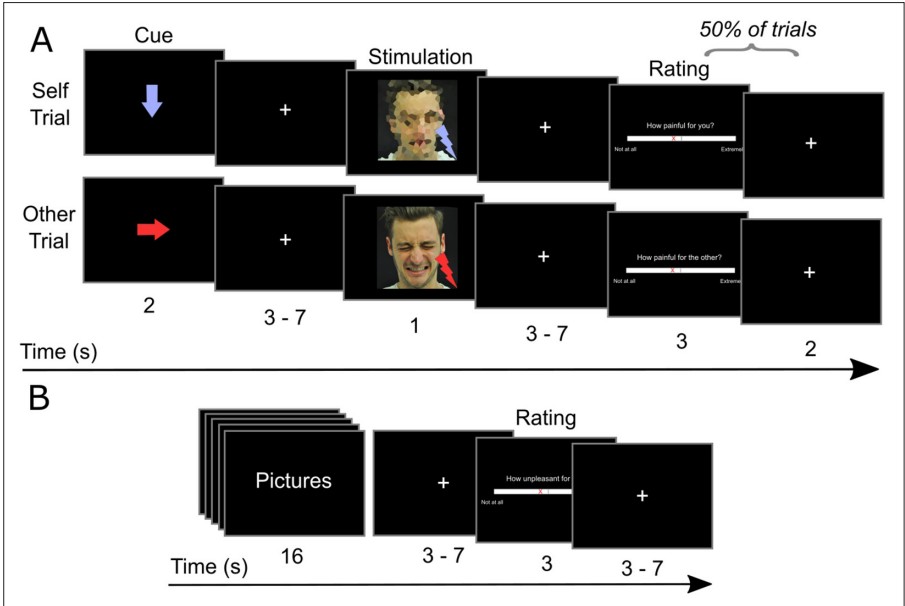

**Figure 1.** Schematic depiction of the experimental tasks. (**A**) Empathy-for-pain task. In trials of the Self condition, participants passively received electrical stimuli. In the Other condition, participants experienced how another person (a confederate) received electrical stimuli. The stimuli were either painful or not painful. In the cue phase, an arrow indicated the recipient (downwards: Self; right: Other) and the intensity (blue: not painful; red: painful) of the next stimulus. In the stimulation phase, the stimulus was delivered. After half of the trials, participants were asked to rate the last stimulus. The confederate depicted has given informed consent that his photograph can be published. (**B**) Emotional reactivity task. Participants were presented pictures with different content (violent or neutral) and different context (real or game context). After observing a block of pictures, participants rated their current unpleasantness on a visual analog scale from 0 to 100.

the many conflicting results reported by experimental research and even meta-analyses (*de Vrieze, 2018*; *Mathur and VanderWeele, 2019*), clearly differentiating between 'absence of evidence' and 'evidence of absence' is particularly important.

To test possible causal effects of VVGs on empathy and its neural correlates, we conducted an experimental prospective study, which addressed each of these limitations. Eighty-nine male participants with little to no prior VVG experience repeatedly played a modified version of *Grand Theft Auto V* over the course of 2 weeks. Participants in the experimental group played a highly violent version of the game and were tasked to kill as many other characters as possible. Participants of the control group played a version of the same game from which all violent content was removed, and were asked to perform a non-violent task (taking photographs of other characters). Before and after this gaming period, participants completed a functional magnetic resonance imaging (fMRI) session during which we measured the behavioral and neural correlates of empathy for pain and emotional reactivity to violent images (see *Figure 1* and Methods: Experimental fMRI sessions for details). We used Bayesian hypothesis tests to assess whether there were negative effects of VVGs on participants' empathic behavior and neural responses. Hypothesis tests were performed by means of the Bayes factor (BF; *Kass and Raftery, 1995*). We followed the convention to report a BF>3 as evidence for the alternative hypothesis, a BF<1/3 as evidence for the null hypothesis, and a BF in the interval [1/3, 3] as inconclusive evidence for either hypothesis (*Kass and Raftery, 1995*; *Keysers et al., 2020*). We would like to emphasize, though, that the BF provides an easily interpretable continuous quantification of the evidence for and against hypotheses, and that a strict categorization of BFs into evidence for and against hypotheses is not necessary. Our aim was to provide conclusive evidence on the question whether VVGs can desensitize humans to the plight of others or not, within our carefully balanced experimental model.

**Table 1.** Posterior parameter means of models for ratings in the empathy-for-pain task. Dependent variable: empathy ratings (visual analog scale, range: 0–100). Factor codings: Group: control game group = –1, violent game group = 1; Session: first session = –1, second session = 1; Intensity: non-painful = –1, painful = 1. Bayes factors were derived from comparing a model where the respective parameter was unrestricted to a model where it was restricted to zero. †These Bayes factors were derived from comparing a model where the parameter was restricted to be negative to a model where it was restricted to zero (one-sided hypothesis test).

| Fixed effect | $\beta$ | 95% Credible interval | | Bayes factor |
|---|---|---|---|---|
| **A) Painfulness ratings** | | | | |
| *Group* | 0.66 | –1.14 | 2.46 | 0.127 |
| *Session* | 0.42 | –0.31 | 1.16 | 0.072 |
| *Intensity* | 27.86 | 25.32 | 30.29 | >100 |
| *Group*Session* | –0.48 | –1.24 | 0.25 | 0.102 |
| *Group*Intensity* | –1.12 | –3.48 | 1.32 | 0.207 |
| *Session*Intensity* | –0.26 | –1.38 | 0.83 | 0.069 |
| *Group*Session*Intensity* | –0.78 | –1.87 | 0.32 | 0.324† |
| **B) Unpleasantness ratings** | | | | |
| *Group* | –1.13 | –4.83 | 2.56 | 0.251 |
| *Session* | –0.63 | –1.94 | 0.71 | 0.141 |
| *Intensity* | 17.48 | 14.93 | 20.11 | >100 |
| *Group*Session* | –0.93 | –2.33 | 0.40 | 0.254 |
| *Group*Intensity* | –0.95 | –3.63 | 1.68 | 0.197 |
| *Session*Intensity* | –1.20 | –2.12 | –0.30 | 0.996 |
| *Group*Session*Intensity* | –0.45 | –1.36 | 0.49 | 0.130† |

## Results

### Behavioral data

#### Descriptive statistics of gaming behavior

Forty-five participants took part as part of the experimental group, and 44 participants as part of the control group. On average, participants of the experimental group killed 2844.7 characters (SD = 993.9, median = 2820, minimum = 441, maximum = 6815). Participants of the control group took an average of 3055.3 pictures of other characters (SD = 1307.5, median = 3026, minimum = 441, maximum = 6815). Thus, as was the aim of our experimental design, each participant of the experimental group was exposed to a substantial number of violent acts in the video game.

#### Empathy for pain

To test our central hypothesis, we investigated if participants who played the VVG showed decreased empathy for pain on the behavioral level. We analyzed the ratings obtained during the empathy-for-pain task with a hierarchical Bayesian censored regression model. We modeled fixed effects for the experimental factors *Group* (non-violent vs. violent gaming, coded as –1 and 1), *Time* (pre vs. post gaming sessions, coded as –1 and 1), and *Intensity* (non-painful vs. painful stimulation of the confederate, coded as –1 and 1), as well as all interactions between these factors. See Methods: Data analysis for more details.

The posterior means of fixed effect parameters are listed in *Table 1*.A for painfulness ratings, and *Table 1*.B for unpleasantness ratings. As a manipulation check, we first tested whether painful stimuli led to increased painfulness and unpleasantness ratings, compared to non-painful stimuli. For both kinds of ratings, this test revealed very strong evidence (BF>100) for an effect of *intensity*, indicating that our paradigm was able to induce empathic responses in participants (see *Figure 2A and B*). The posterior mean of the regression parameter $\beta$ of the factor *Intensity* was 27.86 for painfulness ratings, and 17.48 for unpleasantness ratings. Given our used factor coding, this means that the average difference in ratings between painful and non-painful stimuli was 2*27.86=55.72 points of the 100-point VAS for painfulness ratings, and 2*17.48=34.96 points for unpleasantness ratings.

We found evidence for the absence of a VVG effect on the painfulness ratings. Comparing a model where the fixed effect of *Group*Time*Intensity* could be negative to a model where the effect was set to zero resulted in a BF of 0.324. This means that the observed ratings were about 3.1 times more likely under the null hypothesis of no VVG effect than under the alternative hypothesis. When estimated without restrictions, the posterior mean of $\beta$ for the interaction *Group*Session*Intensity* was –0.78. Given our factor codings, this means that the quantity $[\text{rating}_{Pain} - \text{rating}_{No\ Pain}]_{Session\ 2} - [\text{rating}_{Pain} - \text{rating}_{No\ Pain}]_{Session\ 1}$ (thus, the baseline-corrected empathic response) was on average 1.56 points smaller in the experimental group than in the control group, on the 100-point VAS. However, note that the Bayesian hypothesis test suggests that a model with this interaction restricted to zero provides a better explanation of the data.

For the unpleasantness ratings, evidence for absence of a VVG effect was substantial. With a BF of 0.130, the observed data were about 7.7 times more likely under the null hypothesis of no VVG effect than under the alternative hypothesis. The posterior mean of $\beta$ for the interaction *Group*Session*Intensity* was –0.45. Given our factor codings, this means that the quantity $[\text{rating}_{Pain} - \text{rating}_{No\ Pain}]_{Session\ 2} - [\text{rating}_{Pain} - \text{rating}_{No\ Pain}]_{Session\ 1}$ was on average 0.9 points smaller in the experimental group than in the control group. However, note again that the Bayesian hypothesis test suggests that a model without this interaction provides a better explanation of the data.

In summary, the behavioral data suggest that VVG play as implemented in this study has no effect on either type of empathy rating.

#### Emotional reactivity

Next, we investigated whether playing the VVG desensitized participants toward depictions of violence. We again used a hierarchical Bayesian censored regression model, and included fixed effects for the experimental factors *Group* (non-violent vs. violent gaming, coded as –1 and 1), *Content* (neutral vs. violent, coded as –1 and 1), and *Context* (real vs. game, coded as –1 and 1).

The posterior means of fixed effect parameters of this model are listed in *Table 2*. As a manipulation check, we first tested whether participants experienced more unpleasantness in the emotional

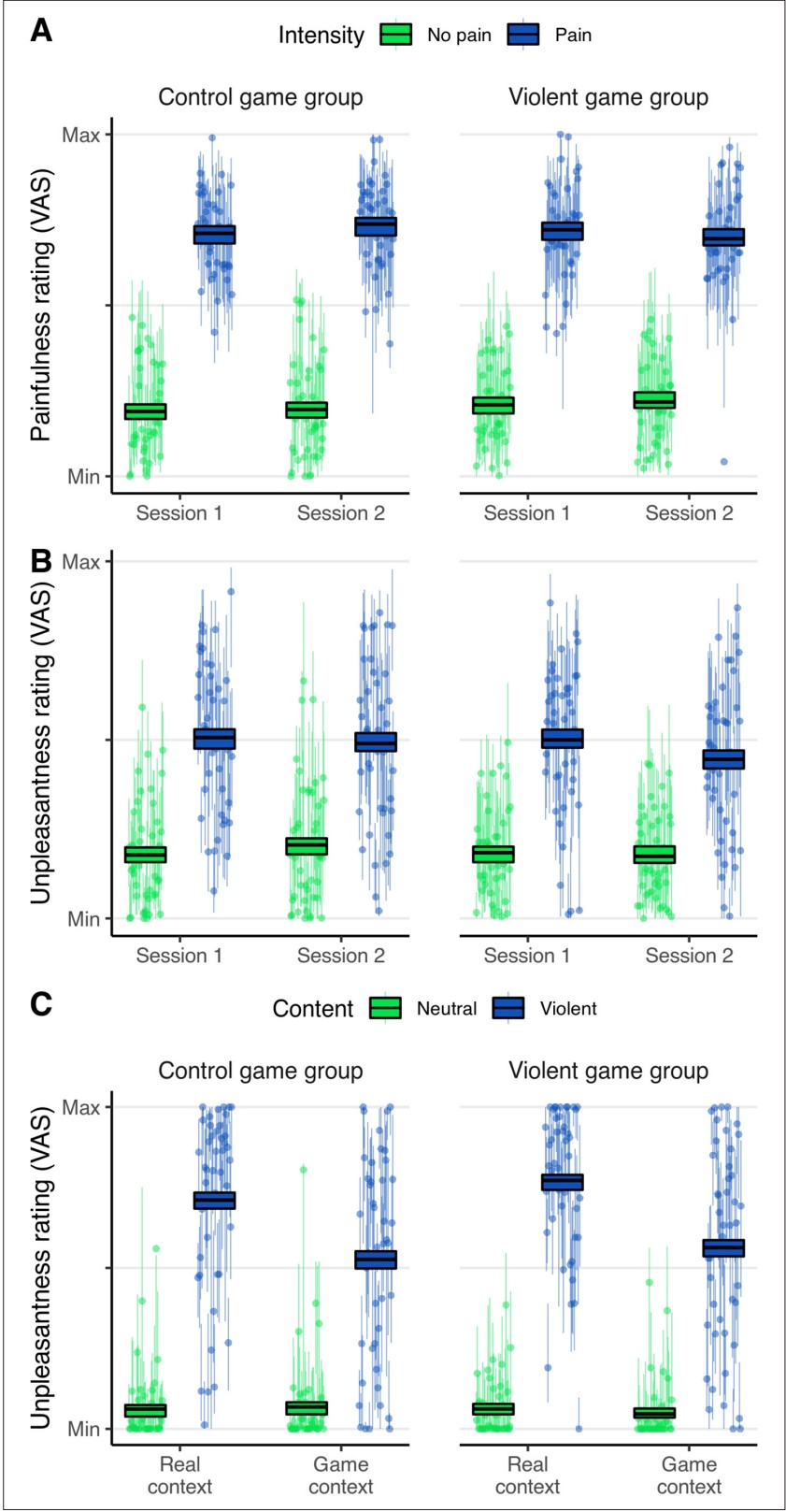

**Figure 2.** Behavioral results. Depicted are participants' ratings during the empathy-for-pain task (**A** and **B**) and the emotional reactivity task (**C**). Ratings were given on a visual analog scale (range: 0–100). (**A**) Empathy for pain, painfulness rating. Question text: 'How painful for the other?'. (**B**) Empathy for pain, unpleasantness rating. Question text: 'How unpleasant for yourself?'. Note that an apparent trend toward a three-way interaction

*Figure 2 continued on next page*

*Figure 2 continued*

Group\*Session\*Intensity is not supported by the respective Bayesian hypothesis test (BF = 0.130, *Table 1*). (**C**) Emotional reactivity, unpleasantness rating. Question text: 'How unpleasant?'. Boxes: the middle line marks the group mean of participant ratings in the respective condition; the box represents the 95% credible interval of the posterior predictive distribution of mean ratings. Dots depict the individual mean ratings of participants, lines depict the 95% credible interval of the posterior predictive distribution of mean ratings of single participants. Control game group: N = 44. Violent game group: N = 45.

reactivity task while observing violent pictures compared to neutral pictures. We found very strong evidence (BF>100) for this hypothesis, indicating that our paradigm was successful in inducing unpleasantness by violent imagery. The posterior mean of the regression parameter $\beta$ of the factor *Content* was 37.08. This means that the average difference in ratings between violent and neutral stimuli was 74.16 points of the 100-point VAS. The unpleasantness ratings are depicted in *Figure 2C*.

Further, we found substantial evidence for the absence of a desensitizing VVG effect. Comparing a model where the fixed effect of *Group\*Content* could be negative to a model where the effect was set to zero resulted in a BF of 0.151. Thus, participants of the violent game group did not show a decreased emotional response toward depictions of real and game violence. Moreover, testing the fixed effect of *Group\*Content\*Context* resulted in a BF of 0.094, indicating that there was also no desensitizing effect that was specific to depictions of game violence. When estimated without restrictions, the regression parameters associated with both interactions were positive, $\beta$=2.28 for *Group\*Content,* and $\beta$=0.33 for *Group\*Content\*Context*. This means that, ostensibly, participants in the experimental group had a very weak tendency to rate violent images as more unpleasant than participants in the control group, contrary to expectations. However, note again that the Bayesian hypothesis test suggests that a model without these interactions provides a better explanation of the data. In summary, the behavioral data suggest that playing the VVG did not emotionally desensitize participants toward violent images.

## fMRI data
### Empathy for pain

We next analyzed the fMRI data collected during the empathy-for-pain task. To define our regions of interest (ROIs), we first performed whole-brain general linear model (GLM) analysis of the data of the first fMRI session. Our contrast of interest [*Other Pain – Other No Pain*] compared brain activity when the confederate experienced painful stimulation to activity when the confederate experienced only non-painful stimulation (see Methods: Data analysis for details). This revealed significant clusters in our a priori defined brain areas of interest, aMCC and bilateral AI, as well as in other areas, including the left supramarginal gyrus and the right angular gyrus (see *Figure 3A*, and Appendix 3 for detailed

**Table 2.** Posterior parameter estimates of models for ratings in the emotional reactivity task. Dependent variable: unpleasantness ratings (visual analog scale, range: 0–100). Factor codings: Group: control game group = –1, violent game group = 1; Content: neutral = –1, violent = 1; Context: real = –1, game = 1. Bayes factors were derived from comparing a model where the respective parameter was unrestricted to a model where it was restricted to zero. †These Bayes factors were derived from comparing a model where the parameter was restricted to be negative to a model where it was restricted to zero (one-sided hypothesis test).

| Fixed effect | $\beta$ | 95% Credible interval | | Bayes factor |
| --- | --- | --- | --- | --- |
| *Group* | 1.26 | –2.78 | 5.44 | 0.349 |
| *Content* | 37.08 | 32.98 | 41.48 | >100 |
| *Context* | –7.24 | –9.15 | –5.39 | >100 |
| *Group\*Content* | 2.28 | –1.92 | 6.52 | 0.151† |
| *Group\*Context* | –1.23 | –3.01 | 0.47 | 0.306 |
| *Content\*Context* | –5.36 | –7.39 | –3.34 | >100 |
| *Group\*Content\*Context* | 0.33 | –1.58 | 2.10 | 0.094† |

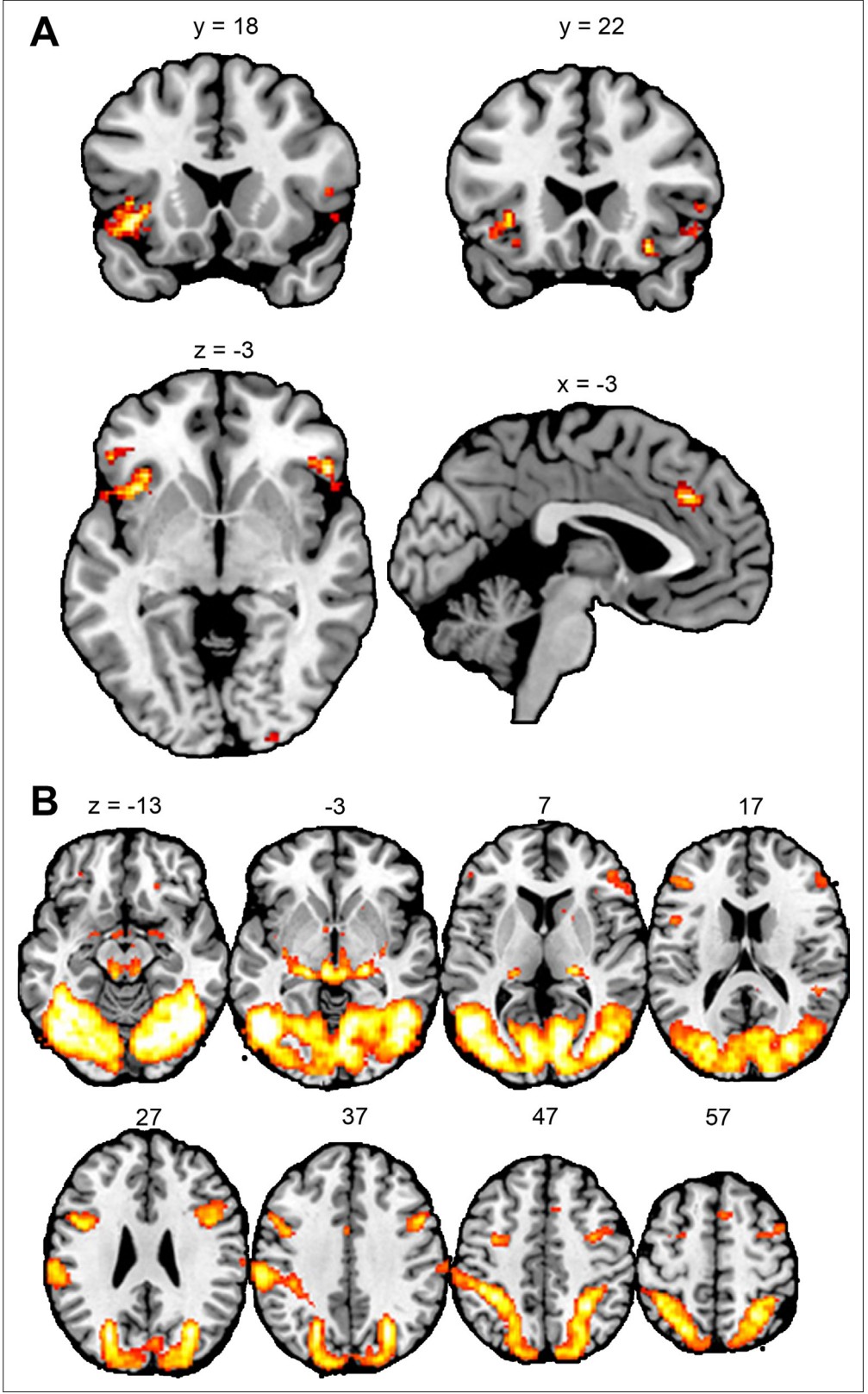

**Figure 3.** Results of the whole-brain analyses for region-of-interest definition. (**A**) Empathy-for-pain task. Clusters represent areas where brain activity was increased when the confederate received a painful electrical stimulus, compared to a non-painful stimulus. (**B**) Emotional reactivity task. Clusters represent areas where brain activity was increased during the observation of violent images, compared to neutral images. All results p<0.05 FWE-corrected. This figure was made with the software MRICron (https://www.nitrc.org/projects/mricron).

results). Subsequently, we performed Bayesian linear mixed effects analyses on the data extracted from the ROIs (aMCC, left AI, right AI). See Methods: Data analysis for details. We compared models where the fixed effect of *Group*Time*Intensity* could be negative to a model where the effect was set to zero. For responses in the *Cue* phase (where participants were informed whether the other person would receive a painful or a non-painful stimulus), we obtained the following BFs: $BF_{aMCC}$ = 0.402; $BF_{left AI}$ = 0.547; $BF_{right AI}$ = 0.190. For responses in the *Stimulation* phase (where participants observed the other person receiving the stimulus), we obtained the following BFs: $BF_{aMCC}$ = 0.176; $BF_{left AI}$ = 0.143; $BF_{right AI}$ = 0.434. See *Appendix 2—table 1* for posterior distributions and BFs of all model parameters. In summary, we found weak to moderate evidence for the absence of an effect of playing the VVG on participants' brain activity while they observed another person in pain.

## Emotional reactivity

Our next analysis concerned the fMRI data coming from the emotional reactivity task. To define our ROIs, we computed the contrast [*Violent – Neutral*], comparing brain activity during observation of violent images to brain activity during observation of images with neutral content (see Methods: Data analysis for details). This revealed significant clusters in one of our a priori areas of interest, the bilateral amygdala, as well as several other regions, such as the bilateral fusiform gyrus and the bilateral precentral gyrus (see *Figure 3B* and Appendix 3 for detailed results). However, we found no significant clusters in the other brain ROIs, the aMCC or the bilateral AI. Therefore, we restricted our subsequent ROI analysis to the amygdala.

We performed Bayesian linear mixed effects analyses on the data extracted from the amygdala. See Methods: Data analysis for details. First, we compared a model where the fixed effect of *Group*-Content* could be negative to a model where the effect was set to zero. This resulted in a BF of 0.324 for the left amygdala, and a BF of 0.338 for the right amygdala, indicating absence of an effect in both ROIs. Next, we tested the fixed effect of *Group*Content*Context*. With a BF of 0.205 for the left amygdala, and 0.163 for the right amygdala, this analysis also indicated the absence of an effect. See *Appendix 2—table 2* for posterior distributions and BFs of all model parameters. In summary, the data suggest that playing the VVG did not lead to a dampened brain response to images of violence in neither real nor gaming contexts.

## Post hoc analyses

### Sample comparability

We constrained our sample to young adult (18–35 years) males who had minimal prior exposure to VVGs in general, and who had not played the game used in the study before. However, given the great popularity of VVGs among young adult males, it is also possible that this constrained our sample to a subpopulation that is less susceptible to desensitization effects to begin with. Therefore, we tested whether the subpopulation from which we drew our sample exhibited higher levels of trait empathy than the general population. To achieve this, we compared the trait empathy levels of our sample, as measured by the Questionnaire for Cognitive and Affective Empathy (QCAE; *Reniers et al., 2011*), to

**Table 3.** Comparison of trait empathy levels between experimental group and control group.
Bayes factors were derived from comparing a model where the mean difference violent video game (VVG) group – Control group was positive to a model where it was restricted to zero (one-sided Bayesian *t*-test).

| QCAE subdimension | VVG group (*N*=83) | | Control group (*N*=132) | | | |
|---|---|---|---|---|---|---|
| | Mean | SD | Mean | SD | t | Bayes factor |
| *Perspective Taking* | 1.93 | 0.43 | 2.01 | 0.51 | –1.189 | 0.074 |
| *Online Simulation* | 1.96 | 0.41 | 1.92 | 0.47 | 0.588 | 0.259 |
| *Emotional Contagion* | 1.58 | 0.48 | 1.63 | 0.55 | –0.659 | 0.098 |
| *Peripheral Responsivity* | 1.58 | 0.58 | 1.58 | 0.62 | 0.021 | 0.155 |
| *Proximal Responsivity* | 1.67 | 0.58 | 1.79 | 0.53 | –1.538 | 0.063 |

those of a control sample of 18- to 35-year-old males who were not preselected for minimal VVG use. See Methods: Data analysis: Post hoc analyses for more details.

The results are depicted in *Table 3*. For all subdimensions, Bayesian *t*-tests provided moderate to substantial evidence for the hypothesis that there is no difference between the two groups (BF<1/3). Thus, our exploratory analysis suggests that our inclusion criterion of minimal VVG exposure did not result in a preselection of individuals with extraordinarily high levels of empathy.

## Test-retest reliabilities

In this study, we measured a number of behavioral and neural correlates in two experimental sessions – once before the exposure to the VVG or the control game, once after. Thus, the test-retest reliability (i.e. the correlation between the two measurements of a variable) is of interest, as this informs us about the relative stability of our outcome variables of interest. This also affects the statistical power of our performed tests (see next section).

For analysis details, see Methods: Data analysis: Post hoc analyses. We found that the test-retest reliability of our behavioral measures of empathy (i.e. participants' ratings) was high to very high (painfulness ratings: $\rho$=0.768, 95% credible interval = [0.613, 0.879]; unpleasantness ratings: $\rho$=0.905, 95% credible interval = [0.813, 0.967]). However, we observed very low test-retest reliability for our neural measurements of empathy (aMCC signal: $\rho$=–0.013, 95% credible interval = [–0.420, 0.402]; left AI signal, $\rho$=–0.001, 95% credible interval = [–0.423, 0.414]; right AI signal, $\rho$=0.027, 95% credible interval = [–0.377, 0.416]).

## Bayesian design analyses

We based our sample size on the results of a power analysis designed for the frequentist inference framework (see section Methods: Power analysis). However, as we ultimately based our inference on BF tests, the theoretical long-term behavior of these tests, given our sample size and expected effect size, is of interest. This also informs us about the effect sizes that could realistically have been detected using our sample size. Therefore, we conducted a post hoc BF design analysis by means of a Monte Carlo simulation experiment (*Schönbrodt and Wagenmakers, 2018*). See Methods: Data analysis: Post hoc analyses for analysis details. It is of particular importance to note that the diagnosticity of hypothesis tests involving repeated measurements does also depend on the correlation between the repeated measures, i.e., the test-retest reliability.

The results are presented in *Table 4*. In summary, the simulation experiment suggested that our behavioral analyses, for which test-retest reliabilities were high, were well enough powered to differentiate between the absence and presence of a medium-to-small effect of *d*=0.3. Note that this effect size is smaller than the lower bound of effect size estimates reported in the meta-analysis of *Anderson et al., 2010*, which was *d*=0.345. For smaller effects, such as *d*=0.2, the a priori power of our behavioral analyses was not optimal, as it would have been likely that we would have obtained an

**Table 4.** Results of the Bayes factor design analysis.

Depicted are the estimated probabilities of inferential decisions for each dependent variable and assumed true effect size *d*. Inc.: Inconclusive evidence, no decision. H0: evidence for the null hypothesis. H1: evidence for the alternative hypothesis. $\rho$=correlation between repeated measurements, i.e., test-retest reliability. The estimated probabilities of correct decisions (evidence for H0 when *d*=0.0, evidence for H1 when *d*>0.0) are marked in bold.

| | Empathy | | | | | | | | | Emotional reactivity | | |
|---|---|---|---|---|---|---|---|---|---|---|---|---|
| | *Painfulness ratings* | | | *Unpleasantness ratings* | | | *Neural response* | | | *Unpleasantness ratings* | | |
| | *($\rho$=0.75)* | | | *($\rho$=0.90)* | | | *($\rho$=0)* | | | *(only second session)* | | |
| **Effect size** | Inc. | H0 | H1 | Inc. | H0 | H1 | Inc. | H0 | H1 | Inc. | H0 | H1 |
| *d*=0.0 | 0.29 | **0.69** | 0.02 | 0.29 | **0.69** | 0.02 | 0.30 | **0.69** | 0.02 | 0.30 | **0.68** | 0.02 |
| *d*=0.2 | 0.58 | 0.20 | **0.23** | 0.44 | 0.05 | **0.50** | 0.49 | 0.43 | **0.08** | 0.55 | 0.33 | **0.13** |
| *d*=0.3 | 0.48 | 0.06 | **0.46** | 0.14 | 0.00 | **0.85** | 0.56 | 0.30 | **0.14** | 0.57 | 0.18 | **0.25** |
| *d*=0.4 | 0.28 | 0.02 | **0.71** | 0.02 | 0.00 | **0.98** | 0.57 | 0.21 | **0.22** | 0.50 | 0.08 | **0.42** |

**Table 5.** Cross-task correlations.

Above diagonal: posterior means of correlations. Below diagonal: 95% credible intervals of correlations. Unpl.=unpleasantness. Emo. Reac.=emotional reactivity. aMCC = anterior midcingulate cortex. AI = anterior insula. Amy = amygdala. l.=left. r.=right.

| | | Emp: Pain | Emp: Unpl. | ER: Unpl. | Emp: aMCC | Emp: lAI | Emp: rAI | ER: lAmy | ER: rAmy |
|---|---|---|---|---|---|---|---|---|---|
| | Empathy: Pain | | 0.630 | 0.280 | 0.015 | 0.043 | 0.133 | –0.037 | –0.037 |
| | Empathy: Unpl. | (0.544,0.708) | | 0.227 | 0.039 | 0.084 | 0.211 | 0.058 | 0.096 |
| *Behavior* | Emo. Reac.: Unpl. | (0.191,0.366) | (0.130,0.323) | | –0.010 | 0.028 | –0.028 | 0.060 | 0.043 |
| | Empathy: aMCC | (–0.178,0.215) | (–0.170,0.245) | (–0.212,0.190) | | 0.055 | 0.069 | 0.040 | 0.050 |
| | Empathy: l. AI | (–0.151,0.235) | (–0.118,0.276) | (–0.167,0.214) | (–0.201,0.285) | | 0.104 | 0.019 | 0.021 |
| | Empathy: r. AI | (–0.056,0.310) | (–0.009,0.393) | (–0.205,0.153) | (–0.156,0.289) | (–0.146,0.323) | | 0.080 | 0.085 |
| | Emo. Reac.: l. Amy | (–0.178,0.096) | (–0.079,0.195) | (–0.080,0.202) | (–0.164,0.241) | (–0.172,0.213) | (–0.124,0.266) | | 0.507 |
| *Neural* | Emo. Reac.: r. Amy | (–0.173,0.098) | (–0.042,0.230) | (–0.095,0.180) | (–0.150,0.251) | (–0.171,0.207) | (–0.117,0.271) | (0.370,0.628) | |

inconclusive result (1/3<BF<3) even in the presence of a true effect of that size. However, given that we obtained evidence for the null hypothesis (BF<1/3) in all relevant BF tests on our behavioral data, our results speak strongly against the presence of such an effect.

Regarding our neural analyses, given the low correlation between repeated measurements (i.e. test-retest reliability), the Bayesian power of our fMRI analyses should be regarded as low. Taken alone, we would not consider them convincing evidence against the presence of a VVG effect. However, together with our behavioral results, they suggest that VVG effects, if they exist, can be expected to be very small.

## Cross-task correlations

Given that we measured empathy for pain and emotional reactivity in the same subjects, our data also allowed us to investigate the relationships between these two phenomena. For this, we calculated the correlations between the behavioral and neural measurements of our outcome variables. The results are presented in *Table 5*. We can observe that for our behavioral measures, cross-task correlations were substantial (*r*=0.227 –0.280, with all credible intervals not covering zero). However, we could observe no substantial cross-task correlations for our neural measures, or across neural and behavioral indicators.

## Discussion

Influential theories of media violence predict that the repeated playing of VVGs results in decreased empathy for pain due to a desensitization to real-world violence (*Anderson et al., 2010*; *Bushman and Anderson, 2002*). Here, we report evidence against this hypothesis in relation to our specific setting. We found that participants who repeatedly played a highly violent game for 7 hr over the course of 2 weeks did not show decreased empathy for another person's pain or decreased responses to violent imagery.

Our findings contrast with several earlier studies that found a negative relationship between playing VVGs and empathic responses to violence. Importantly, the majority of these studies were quasi-experimental in nature, and therefore provide only limited evidence for a putative causal effect of violent gaming (*Bartholow et al., 2005*; *Bartholow et al., 2006*; *Gentile et al., 2016*; *Krahé et al., 2011*). Moreover, the few experimental studies that exist implemented designs investigating short-term carryover effects, as they had exposed participants to virtual violence rather immediately before measurements of their outcome variables of interest (*Arriaga et al., 2011*; *Bushman and Anderson, 2009*; *Carnagey et al., 2007*; *Engelhardt et al., 2011*; *Guo et al., 2013*; *Staude-Müller et al., 2008*). Together with the study of *Kühn et al., 2018*, our study is one of the first to investigate persistent effects of VVGs in participants without prior experience with them, enabling a clear assessment of the causality of VVG effects. Importantly, our study was designed to address several limitations of the study of Kühn et al.: We strictly controlled the amount of virtual violence actually experienced by participants, and used a non-violent version of the same game

in the control condition; moreover, we applied a Bayesian analytical approach, which, together with our comparatively large sample size, enabled us from the outset to distinguish 'absence of evidence' from 'evidence of absence' of VVG effects. This approach yields consistent evidence from both behavioral and neural data that VVGs, to the extent and characteristics played in our interventional design, are not causally responsible for a persistent lack of empathy or emotional desensitization to violence.

Despite the aforementioned strengths of our study, we also need to address several limitations. Our experimental design ensured that participants of the experimental group were exposed to a substantial amount of violent gameplay during gaming sessions (each participant 'killed' an average of 2845 other characters in a graphically violent way). However, the overall exposure to virtual violence was still very low when compared to the amount that is possible in the everyday life of typical VVG players. During our experiment, participants played for 7 hr over the course of 2 weeks. However, habitual gamers can play an average of 16 hr in the same time frame (*Clement, 2021*; *Statista Research Department, 2022*). Our results cannot preclude that longer and more intense exposure to VVGs could have negative causal effects on empathy. In particular, adolescents and children as well as persons with specific neuropsychiatric traits might be especially susceptible to long-term changes due to increased brain plasticity. However, empirically testing higher levels of violence with the same degree of control as realized in our study would reach the limits of practical feasibility. We thus believe that our results provide an important perspective on the size of VVG effects that could realistically be expected in experimental research.

To increase experimental control, we restricted our sample to young adult males who had minimal prior exposure to VVGs. It is possible that, due to this strict preselection criterion, our sample was drawn from a subpopulation that is particularly resistant to desensitization. An exploratory analysis provided strong evidence that our selection criterion did not result in particularly high levels of trait empathy in our sample, though. However, we cannot preclude that our sample was particularly resistant to VVG effects due to other, untested characteristics. Further research is needed to assess if our results generalize to samples with other characteristics that may be more representative for the general population.

To maximize the amount of violence that participants would be exposed to (and commit) in the game, we restricted the game's objective to killing other characters, and incentivized this behavior with monetary rewards. This might have reduced the ecological validity of our operationalization of gaming, and it is possible that bigger effects could be seen when violent gameplay is more internally motivated, i.e., individuals who want to play the game may be differently affected than those that have merely accepted to be part of an experiment. Still, our results provide valid evidence that the mere exposure to virtual violence for 7 hr over 2 weeks is not sufficient to decrease empathy.

It should be noted that there are few studies that connect laboratory-based experimental investigations of empathy and emotional reactivity to real-world behavior and its measures. There are indications, however, that neuroscientifc empathy measures similar to the ones used here predict individual social behavior (e.g. donation, helping, or care-based behavior; *Ashar et al., 2017*; *Hartmann et al., 2022*; *Hein et al., 2010*; *Tomova et al., 2017*), and that they are also validated by their predictivity of mental or preclinical disorders characterized by deficits in empathy (*Bird et al., 2010*; *Lamm et al., 2016*, for review). That said, it is obvious that future research is needed that bridges and integrates laboratory and field-based measures and approaches, in order to inform us how changes (or their absence) in neural responses induced by VVG play are connected to real-life social emotions and behaviors (see *Stijovic et al., 2023*, for a recent example illustrating, in the domain of social isolation research, how a combined lab- and field-based study can be directly informed by prior laboratory-based neuroscience findings).

Our study was designed to reliably detect an effect size of $d$=0.3, an effect even smaller than the lower estimate for VVG effects on empathy reported in *Anderson et al., 2010*. Our results provide substantial evidence that effects of this magnitude are not present in settings similar to our experimental design. These arguments notwithstanding, it needs to be noted that future studies with higher power may detect still smaller effects. Considering the high prevalence of VVG, even such small effects could be of high societal relevance (*Funder and Ozer, 2019*). For now, based on the current design and data, we can conclude that experimental long-term VVG effects on empathy are unlikely to be as large as previously reported.

It may be argued that the empathy for pain paradigm and the associated behavioral and neural responses are so robust and resistant to changes by external factors that this may explain the lack of evidence for the effects of VVG play. This argument however would contradict a wealth of findings illustrating malleability of empathic responses using this and related designs, including with placebo analgesia (*Rütgen et al., 2021*; *Rütgen et al., 2015a*; *Rütgen et al., 2015b*), an intervention that usually shows low to moderate effect sizes as well (see e.g. *Hein and Singer, 2008*; *Jauniaux et al., 2019*; *Lamm et al., 2019*, for review).

Lastly, and somewhat surprisingly, we found that the test-retest reliability of our neural covariates of empathy for pain were close to zero for all investigated ROIs. Knowing that an individual's neural empathic response (blood oxygen level-dependent [BOLD] activity for seeing somebody else in pain vs. in no pain) was above or below average in the first session provides little to no information about their relative response in the second session. To the best of our knowledge, our study was the first one to present the empathy for pain paradigm to the same sample of participants after a longer time frame. Thus, this surprising result provides valuable information on the limitations of this task respectively the neural measurements acquired in it, and certainly demands further research to investigate the factors influencing fMRI reliability (see also *Elliott et al., 2020*; *Kragel et al., 2021*). We would like to emphasize, though, that a high test-retest reliability is not a precondition for the valid testing of group-level effects. For a group-level effect to be testable, it is only necessary that the mean of the dependent variable is consistently affected by the independent variable. It is not necessary that participants who show an above average level in the DV in one session also show an above average level in the second session, and vice versa. Otherwise, there would also be no point in independent-sample designs. Indeed, it has recently been discussed that highly robust cognitive tasks are bound to exhibit low test-retest reliability, as robust tasks are often characterized by low interindividual variation, and thus leave only little variance that can be explained by participant traits (*Hedge et al., 2018*). However, it must also be noted that low reliability does lead to lower power of repeated measures designs. As discussed above, the low reliability of the measured neural responses has resulted in suboptimal power of our tests on fMRI data.

In summary, our findings stand in contrast to claims that posit the playing of violent games as an essential factor for explaining decreases in empathy. If this is shown to generalize to when people play more often and over longer periods, the desensitization to violence described in prior reports using quasi-experimental designs might have been caused by third and pre-existing factors, such as education, socio-economic status, or mental health issues (*DeCamp and Ferguson, 2017*; *Lemmens et al., 2006*; *Shao and Wang, 2019*; *Tortolero et al., 2014*). Together with similar findings (*Kühn et al., 2018*), our results point out the limits to which VVGs can be held responsible for lacks of empathy, at least in highly controlled experimental settings that last for the 2 weeks of play implemented here. This is not to say, though, that there is no point in further investigating the complex relationships between violent media use and adverse social behavior. We propose that the design and analysis approach of the present study could act a reference of how future studies should be conducted, in order to increase the stringency and robustness of research in this domain. Together with our findings, such studies will aid in resolving the scientific controversy regarding the negative effects of VVGs (*de Vrieze, 2018*; *Mathur and VanderWeele, 2019*), and contribute to a deeper understanding of the interplay between violent media and emotion.

## Methods

### Power analysis

We planned to collect data from 90 participants. We derived this sample size from a power analysis based on VVG effect sizes reported in the meta-analysis of *Anderson et al., 2010*. The authors estimated the size of the negative VVG effect on empathy/desensitization to be $r=0.194$, 95% CI = [0.170, 0.217], which corresponds to Cohen's $d=0.396$, 95% CI = [0.345, 0.445], representing a small-to-medium effect. We chose $d=0.300$ as the minimum effect size for which we wanted to achieve a power of 0.80, to ensure that we would have enough power even if the reported effect size was overestimated. Note that thus, the effect size we used was even smaller than the lower bound reported in *Anderson et al., 2010*. We performed the power analysis using the software Gpower 3.1.9.2 (*Faul et al., 2007*), calculating the required sample size to achieve a power of 0.8 for the interaction in

a 2-by-2 within-between design ANOVA, assuming a medium correlation of 0.5 between repeated measures, and using the conventional alpha error level of 0.05. This resulted in a required sample size of 90. Using such a sample size, the achieved power for the effect size reported in *Anderson et al., 2010*, as well as its lower and upper bound, was as follows: for $d=0.345$, achieved power = 0.901; for $d=0.396$, achieved power = 0.960; for $d=0.445$, achieved power = 0.986.

Please note that while this power analysis was based on a frequentist analysis framework, we are reporting Bayesian analyses here. However, we considered this power analysis to be a sensible benchmark for the sample size needed to answer our research questions. See Results: Post hoc analyses: Bayesian design analysis for a Bayesian design analysis that provides more information on the size of effects that could be detected with our sample size using Bayesian analyses.

## Participants

In total, 97 participants completed the first experimental session. Of these, eight participants dropped out of the study (six before the first video game sessions; two after, of which one was from the experimental group and one from the control group). We thus acquired complete datasets from 89 participants.

To control for previous VVG exposure, we only included individuals that had not played VVGs at least 12 months before testing, and had not played the video game *Grand Theft Auto V* before. We did this to avoid a possible ceiling effect: participants who had already played these games before might already have been desensitized too much for our experimental VVG exposure to show any effect, therefore reducing sensitivity. We tested only male participants, as more males than females play VVGs regularly (*Gentile et al., 2004*; *Krahé and Möller, 2004*; *Padilla-Walker et al., 2010*). Moreover, males have been shown to be more easily influenced by violent media (*Bartholow and Anderson, 2002*; *Bettencourt and Kernahan, 1997*). To further increase homogeneity of the sample, we restricted the age range of possible participants to 18–35 years. Additional inclusion criteria were no history of neurological or psychiatric disorders or drug abuse, and standard inclusion criteria for MRI measurements. Participants were recruited through online advertisements and received a financial compensation of €145 for participating in all experimental sessions. A performance-linked bonus of up to €35 acted as an additional incentive during the game sessions. The study was approved by the ethics committee of the Medical University of Vienna (decision number 1258/2017). The confederate depicted in *Figure 1A* has given informed consent that his photograph may be used for this publication.

## Overall study design

Participants were randomly assigned to the violent game group or the control game group. Participants first completed a pretest fMRI session, during which they performed an experimental task designed to measure empathy for pain. Then, over the course of 2 weeks, participants of the violent game group repeatedly played a VVG, while the control game group played a non-violent version of the same game. Subsequently, both groups completed the posttest fMRI session. Here, participants performed the empathy-for-pain paradigm again, and also completed a task designed to measure emotional reactivity to violent pictures.

## Experimental fMRI sessions

### Confederate

To facilitate empathic responses during the experimental tasks, participants completed the experimental session together with a male confederate. The confederate acted as if he were a second participant of the experiment. This deception was maintained until the end of the last experimental session, at which point participants were debriefed.

### Pain calibration

The empathy-for-pain paradigm included the administration of painful but tolerable stimuli. The physical pain was induced via a well-established procedure (e.g. *Rütgen et al., 2015b*). Electrical stimuli were produced by a Digitimer DS5 stimulator (Digitimer Ltd, Clinical & Biomedical Research Instruments, United Kingdom) and delivered by electrodes placed on the dorsum of the left hand. Subjective pain thresholds were determined using a standardized calibration procedure. The participant

received short (500 ms) stimuli of increasing intensity and was asked to rate pain intensity on a numeric scale (0 = 'not perceptible'; 1 = 'perceptible, but not painful', 3 = 'a little painful', 5 = 'moderately painful', 7 = 'very painful', 9 = 'extremely painful, highest tolerable pain'). The average intensities of stimuli rated as 1 and 7 were then chosen as the intensities of the non-painful and painful stimulation conditions during the empathy-for-pain task.

## Empathy-for-pain paradigm

We used a well-established paradigm to measure participants' empathic responses (*Hartmann et al., 2021*; *Rütgen et al., 2015a*; *Rütgen et al., 2015b*; *Singer et al., 2004*). Participants either received electric stimuli themselves (*Self* condition), or saw images of the confederate indicating that he was currently receiving electric stimulation (*Other* condition). The stimuli were either painful (*Pain* condition) or perceptible but not painful (*No Pain* condition). The timeline of the task is illustrated in *Figure 1A*. At the start of each trial, a downwards or rightwards arrow (presented for 2 s) indicated whether the next stimulus would be delivered to the participant or the confederate, respectively (Cue phase). Red and blue arrows indicated painful and non-painful stimulation, respectively. After a jittered interval [3–7 s], the stimulus was delivered (Stimulation phase). In the Self condition, the participant received the electrical stimulus (0.5 s), and saw a pixelated photograph (1 s). In the Other condition, the participant saw a photograph of the confederate with a neutral or painful facial expression. After half of the trials, participants rated the last stimulus on a 100-step visual analog scale (VAS). In the Self condition, participants rated how painful the last stimulus was for themselves. In the Other condition, participants rated how painful the stimulus was for the confederate (other-oriented painfulness rating), and how unpleasant it was for themselves to observe the confederate receiving the stimulus (self-oriented unpleasantness rating). In total, there were 64 trials, with 16 trials per condition (Self Pain, Self No Pain, Other Pain, Other No Pain). Conditions were presented in a pseudorandomized order. The task was presented using COGENT (http://www.vislab.ucl.ac.uk/cogent.php), implemented in MATLAB 2017b (The MathWorks Inc, Natick, MA, USA). The total task duration was approx. 20 min.

## Emotional reactivity paradigm

To investigate emotional reactivity to violent images, we used an affective picture paradigm (*Olofsson et al., 2008*; *Petrovic et al., 2005*). Participants were shown pictures of either neutral or violent content (factor *Content*). Additionally, the pictures depicted either real scenes, or scenes taken from the video game participants played during the gaming sessions (factor *Context*). Real pictures were taken from the International Affective Pictures System (IAPS; *Lang et al., 2005*). Game pictures were matched to IAPS pictures in terms of content, valence, and arousal (see Appendix 1).

The sequence of events of the task is illustrated in *Figure 1B*. Each block consisted of five pictures of the same condition, presented for 3 s each, and with a short interval of 0.2 s between pictures. After a jittered interval [3–7 s] participants rated how unpleasant they felt on a 100-step VAS. In total, participants saw 16 blocks of pictures, with 4 blocks per condition (Neutral Real, Neutral Game, Violent Real, Violent Game). The task was presented using COGENT, and total task duration was approx. 5 min. To avoid that participants formed expectations about the purpose of the study early on, participants completed this task only in the second fMRI session.

## MRI data acquisition

MRI data were acquired with a 3T Siemens Skyra MRI system (Siemens Medical, Erlangen, Germany) and a 32-channel head coil. BOLD functional imaging was performed using a multiband-accelerated echoplanar imaging sequence with the following parameters: Echo time (TE): 34 ms; repetition time (TR): 1200 ms; flip angle: 66°; interleaved ascending acquisition; 52 axial slices coplanar to the connecting line between anterior and posterior commissure; multiband acceleration factor 4, resulting in 13 excitations per TR; field-of-view: 192×192×124.8 mm³, matrix size: 96×96, voxel size: 2×2×2 mm³, interslice gap 0.4 mm. Structural images were acquired using a magnetization-prepared rapid gradient-echo sequence with the following parameters: TE = 2.43 ms; TR = 2300 ms; 208 sagittal slices; field-of-view: 256×256×166 mm³; voxel size: 0.8×0.8×0.8 mm³. To correct functional images for inhomogeneities of the magnetic field, field map images were acquired using a double echo gradient echo sequence with the following parameters: TE1/TE2: 4.92/7.38 ms; TR = 400 ms; flip angle: 60°;

36 axial slices with the same orientation as the functional images; field-of-view: 220×220×138 mm³; matrix size: 128×128×36; voxel size: 1.72×1.72×3.85 mm³.

## Gaming sessions

Between the two fMRI sessions, participants came seven times to the laboratory to play a video game for 1 hr. Intervals between subsequent gaming sessions were approximately 24–48 hr, and the second fMRI session was completed at least 24 hr after the last gaming session. Participants of both groups played a modified version of the game *Grand Theft Auto V*. In the violent game group, participants controlled a male character equipped with a close-combat weapon, and were tasked to kill as many other characters as possible. Killing was graphically violent, as hitting a character was accompanied by the splattering of blood, realistic animations of injury, and screams. In the control game group, participants played a version of the game in which all violence was removed. The player character had no weapon, and could not hurt other characters in any way. They could also not be attacked by other characters, and there was no violence between non-player characters. In this condition, participants were tasked to take photographs of as many other characters as possible. In both groups, participants could also freely explore the world of the game. To incentivize a high number of violent or non-violent acts, each kill or photograph was rewarded with one point. For every two points, participants were paid out +0.01€ at the end of the study.

Due to the lack of other studies implementing a randomized experimental prospective design (except for *Kühn et al., 2018*, published while data collection was already ongoing), there were no benchmarks for the amount and frequency of video game exposure for our study. We chose our regimen (seven 1-hourly sessions over 2 weeks) as we considered this a substantial yet still feasible amount of exposure. Number of sessions, playing time per session, and total playing time were considerably higher than in previous studies reporting VVG effects on empathy (*Arriaga et al., 2011*; *Carnagey et al., 2007*; *Engelhardt et al., 2011*; *Hasan et al., 2013*).

## Data analysis

In this paper, we follow a Bayesian data analysis approach (*Keysers et al., 2020*), which allows clear assessments of the presence or absence of an effect of VVGs on empathy. Hypothesis tests were performed by means of the BF (*Kass and Raftery, 1995*). The BF represents how much more probable the observed data is under the alternative hypothesis compared to the null hypothesis. A well-established convention is to report a BF>3 as evidence for the alternative hypothesis, a BF<1/3 as evidence for the null hypothesis, and a BF in the interval [1/3, 3] as inconclusive evidence for either hypothesis (*Kass and Raftery, 1995*; *Keysers et al., 2020*). We formulated informed priors for all models to enable valid BF hypothesis tests (*Vanpaemel, 2010*). To increase comparability with the results of previous papers, we also report analogous frequentist analyses in the Appendix 5. We registered the analysis plan of this study at https://osf.io/yx423/.

### Behavioral data analysis

To test the effects of VVGs on behavioral measures of empathy for pain, we analyzed the VAS ratings obtained during the empathy-for-pain task with hierarchical Bayesian censored regression models. We used censored regression models to account for the fact that participants could give no ratings lower than 0, or higher than 100. Models were estimated using the R package *brms* (*Bürkner, 2017*). We modeled fixed effects for the experimental factors *Group* (non-violent vs. violent gaming, coded as –1 and 1), *Time* (pre vs. post gaming sessions, coded as –1 and 1), and *Intensity* (non-painful vs. painful stimulation of the confederate, coded as –1 and 1), as well as all interactions between these factors. Additionally, we modeled per-subject random effects of *Time*, *Intensity*, and these factors' interaction term. To further account for variations in how participants used the VAS rating scale, we modeled per-subject error variance terms. For further details about the model specification and prior formulation, see Appendix 2.

We used the same kind of model to test possible desensitizing effects of VVGs on emotional reactivity to violent images. Here, we modeled fixed effects for the experimental factors *Group* (non-violent vs. violent gaming, coded as –1 and 1), *Content* (neutral vs. violent, coded as –1 and 1), and *Context* (real vs. game, coded as –1 and 1). Additionally, we modeled per-subject random effects for *Content*, *Context*, and their interaction, as well as per-subject error variances.

## MRI data preprocessing

Preprocessing and analysis of fMRI data were performed using SPM12 (Wellcome Trust Centre for Neuroimaging, https://www.fil.ion.ucl.ac.uk/spm) implemented in MATLAB 2017b. Functional images were slice timed and referenced to the middle slice, realigned to the mean image, and unwarped using the acquired field map. The structural image was co-registered to the mean image of the realigned functional images using mutual information maximization, and structural and functional images were normalized to the stereotactic Montreal Neurological Institute (MNI) space. The normalized functional images were smoothed with a Gaussian kernel of 4 mm full-width-at-half-maximum, which is equal to twice the voxel size on every axis. To remove motion-related artifacts, the functional images were then subjected to an independent-component-analysis based algorithm for automatic removal of motion artifacts (*Pruim et al., 2015a*; *Pruim et al., 2015b*), implemented using the FMRIB software library (FSL v5.0; http://www.fmrib.ox.ac.uk/fsl).

## fMRI analyses: empathy for pain

With regard to empathy, our central interest lay in modulations of AI and ACC activity. To identify the regions in which empathic responses were reliably elicited independently of our experimental manipulation, we first analyzed the data from the first experimental session. We performed GLM-based whole-brain analysis using SPM12 (Wellcome Trust Centre for Neuroimaging, https://www.fil.ion.ucl.ac.uk/spm), implemented in MATLAB 2017b. For each participant, the design matrix included regressors for the Cue and Stimulation events, separate for all four combinations of conditions (Self No Pain; Self Pain; Other No Pain; Other Pain). As nuisance regressors, we included regressors for the rating events. We then subjected the beta images of the first-level contrast *Other Pain>Other No Pain* to a one-sample *t*-test, and identified the voxels in which this contrast was significant and positive (p<0.05 after family-wise error correction). From this, we obtained a binary mask of significant voxels. We then intersected this mask with anatomical masks taken from the Automated Anatomical Labeling atlas (AAL; *Tzourio-Mazoyer et al., 2002*). For the AI ROI, the binary mask was intersected with the AAL mask of the insula (label IN). For the aMCC ROI, the binary mask was intersected with the AAL masks of the anterior and median cingulate and paracingulate gyri (labels ACIN and MCIN). The aim of this masking procedure was to restrict analyses to those parts of the brain areas that are actually recruited by the task. We believe that this increases the sensitivity of our analyses, as we remove signals from voxels that are also part of these anatomical regions, but not actually recruited by the task.

We analyzed signal changes extracted from our ROIs with Bayesian linear mixed effects model tailored for fMRI data. Note that the ROIs, which were based on the signal from only the first session, were used to extract signals from both sessions. Custom code for this analysis with the software STAN (*Carpenter et al., 2017*) can be found at https://osf.io/yx423/. See also the Appendix 2 for more information. The full model included regressors for the Cue and Stimulation events, as well as nuisance regressors for rating events.

## fMRI analyses: emotional desensitization

When testing the effects of VVGs on brain activity during the emotional-reactivity task, our main interest lay in a possible modulation of responses in the amygdala, as well as aMCC and AI. To define the corresponding ROIs, we first identified the brain areas that were reliably activated by violent imagery, independent of the experimental manipulation, using whole-brain GLM analysis. For each participant, the design matrix included regressors for the blocks of picture presentations, separate for all four combinations of conditions (Neutral Real, Neutral Game, Violent Real, Violent Game). As nuisance regressors, we included regressors for the rating events. We then pooled the beta images of the first-level contrast *Violent>Neutral* across both groups, and subjected them to a one-sample *t*-test. From this, we obtained a binary mask of voxels significant at p<0.05 after family-wise error correction. We then intersected this mask with AAL masks to obtain our final ROIs (for AI: label IN; for aMCC: labels ACIN and MCIN; for amygdala: label AMYG). We analyzed signal changes extracted from our ROIs with Bayesian linear mixed effects model. The full model included regressors for the blocks of picture presentation, as well as nuisance regressors for rating events.

## Post hoc analyses

### Sample comparability

Due to our preselection of young adult males with minimal prior VVG exposure, it appeared possible that our sample was drawn from a subpopulation with higher trait empathy than the general population. To test this potential limitation, we compared the trait empathy levels of our sample, as measured by the QCAE (*Reniers et al., 2011*), to those of a control sample of 18- to 35-year-old males who were not preselected for minimal VVG use. The control sample was taken from the dataset of *Borghi et al., 2023*, which is freely accessible online (https://osf.io/ujp3e). We chose this open dataset because we deemed it highly comparable to our own sample, having also been drawn from the Austrian population, by researchers of the same university. To test whether our sample exhibited higher trait empathy levels than the control sample, we calculated a one-sided Bayesian *t*-test for each of the five subdimensions of the QCAE, using the R package BayesFactor (*Morey and Rouder, 2022*).

### Test-retest reliabilities

Given that our experimental design included measurements of participants' empathic responses in two sessions (once before playing the VVG or the control game, once after), the test-retest reliability $\rho$ of these two measurements was of interest.

In our behavioral data, the empathic response in one session was given by the average difference in ratings for *Pain* trials minus *No Pain* trials in session 1 and 2. Given our estimated hierarchical Bayesian censored regression models, the test-retest reliability of empathic responses can be estimated as

$$\rho = \frac{Cov\left(b_I\, b_{I:\,S},\, b_I + b_{I:\,S}\right)}{\sqrt{Var\left(b_I - b_{I:\,S}\right)\, Var\left(b_I + b_{I:\,S}\right)}}$$

where *Cov* and *Var* are the Covariance and Variance, respectively, $b_I$ is the random effect of the factor *Intensity* (Pain vs. No Pain), and $b_{I:\,S}$ is the random effect of the interaction of factors *Intensity* and *Session*. By the bilinearity of the covariance operator, this formula can be written in terms of estimated model parameters as

$$\rho = \frac{\sigma_{b_I}^2 - \sigma_{b_{I:\,S}}^2}{\sqrt{\left(\sigma_{b_I}^2 + \sigma_{b_{I:\,S}}^2 - 2r_{b_I b_{I:\,S}}\sigma_{b_I}\sigma_{b_{I:\,S}}\right)\left(\sigma_{b_I}^2 + \sigma_{b_{I:\,S}}^2 + 2r_{b_I b_{I:\,S}}\sigma_{b_I}\sigma_{b_{I:\,S}}\right)}}$$

where $\sigma_{b_I}^2$ and $\sigma_{b_{I:\,S}}^2$ are the variances of the random effect of *Intensity* and *Intensity:Session*, respectively, and where $r_{b_I b_{I:\,S}}$ is the correlation between these two random effects.

In our neural data, we defined the empathic response in one session as the average difference in BOLD signal to observing the other in pain vs. observing the other in no pain. Given our estimated hierarchical Bayesian regression model, the test-retest reliability of the neural response was given by the correlation coefficient between the random effect for the regressor *Stimulus Other: Pain – No Pain* in Session 1 and the random effect for the equivalent regressor in Session 2.

### Bayesian design analysis

We based our sample size on the results of a power analysis designed for the frequentist inference framework (see section Methods: Power analysis). However, as we ultimately based our inference on BF tests, the theoretical long-term behavior of these tests, given our sample size and expected effect size, is of interest. Therefore, we conducted a post hoc BF design analysis by means of a Monte Carlo simulation experiment (*Schönbrodt and Wagenmakers, 2018*).

The analysis was performed using the R package *BayesFactor* (*Morey and Rouder, 2022*). We simulated data from the scenario in which there was no VVG effect on the outcome variable (H0; Cohen's *d*=0), as well as from three scenarios where there was a true VVG effect (H1). Here, we considered three different effect sizes: *d*=0.4, which is close to the effect size estimate of *Anderson et al., 2010*; the exact estimate was (*d*=0.394); *d*=0.3, which is the effect size we used in our power analysis; and *d*=0.2, the conventional threshold for small effects.

For each scenario/effect size, we randomly generated 10,000 datasets of the same size as our real sample (control group = 44 participants; experimental group = 45 participants) and subjected them

to BF hypothesis tests, assessing whether the BF provided evidence for the alternative hypothesis (BF>3), for the null hypothesis (BF<1/3), or inconclusive evidence (1/3<BF<3). For the behavioral and neural empathy measures, which were measured in two sessions (once before playing the VVG or the control game, once after), we used test-retest-reliability estimates that are close to those from the previous section.

## Cross-task correlations

We additionally report the empirical correlations between the behavioral and neural measurements of our participants empathic response in the empathy-for-pain task, and their response in the emotional reactivity task. As indicators of participants' behavioral responses, we used their estimated random effects from the Bayesian hierarchical models on their rating data (for Empathy for Pain: factor *Intensity*, i.e. Pain vs. No Pain; for Emotional Reactivity: factor *Context*, i.e. Violent vs. Neutral). As indicators of participants' neural responses, we used their estimated random effects from the Bayesian models on signals extracted from the ROIs (for Empathy for Pain: regressor *Stimulus Other: Pain – No Pain;* for Emotional Reactivity: regressor *Violent – Neutral*).

## Acknowledgements

This work was funded in part by the Vienna Science and Technology Fund (WWTF VRG13-007), a Hjärnfonden (FO2014-0189) grant and a Karolinska Institutet 2015 (2-70/2014-97) grant awarded to PP, and a Knut and Alice Wallenberg Foundation (KAW 2014.0237) grant awarded to AO. We would like to thank Sophia Shea, Leonie Brög, and Johannes Ayrle for assistance during data collection.

## Additional information

### Funding

| Funder | Grant reference number | Author |
|---|---|---|
| Vienna Science and Technology Fund | WWTF VRG13-007 | Claus Lamm |
| Hjärnfonden | FO2014-0189 | Pedrag Petrovic |
| Karolinska Institutet | 2-70/2014-97 | Pedrag Petrovic |
| Knut och Alice Wallenbergs Stiftelse | KAW 2014.0237 | Andreas Olsson |

The funders had no role in study design, data collection and interpretation, or the decision to submit the work for publication.

### Author contributions

Lukas Leopold Lengersdorff, Conceptualization, Data curation, Software, Formal analysis, Investigation, Visualization, Methodology, Writing – original draft, Project administration, Writing – review and editing; Isabella C Wagner, Conceptualization, Supervision, Methodology, Writing – review and editing; Gloria Mittmann, Investigation, Methodology, Project administration; David Sastre-Yagüe, Andre Lüttig, Software, Investigation, Methodology; Andreas Olsson, Pedrag Petrovic, Conceptualization, Supervision, Funding acquisition, Methodology, Writing – review and editing; Claus Lamm, Conceptualization, Resources, Supervision, Funding acquisition, Methodology, Writing – original draft, Writing – review and editing

### Author ORCIDs

Lukas Leopold Lengersdorff ⓘ http://orcid.org/0000-0002-8750-5057
Isabella C Wagner ⓘ http://orcid.org/0000-0002-4383-8204
Gloria Mittmann ⓘ http://orcid.org/0000-0003-2750-7779
Andre Lüttig ⓘ http://orcid.org/0000-0002-6026-6834
Pedrag Petrovic ⓘ http://orcid.org/0000-0002-5536-945X
Claus Lamm ⓘ https://orcid.org/0000-0002-5422-0653

## Ethics

The study was approved by the ethics committee of the Medical University of Vienna (decision number 1258/2017). All participants gave informed consent prior to the start of the first experimental session. The confederate depicted in Figure 1A has given informed consent that his photograph may be used for this publication.

## Decision letter and Author response

Decision letter https://doi.org/10.7554/eLife.84951.sa1
Author response https://doi.org/10.7554/eLife.84951.sa2

---

## Additional files

### Supplementary files

• MDAR checklist

### Data availability

Behavioral data, fMRI signal timecourses extracted from our regions of interest, task event timings, custom STAN code, and game images used in the emotional reactivity task are accessible at Open Science Framework. Unthresholded statistical maps are accessible at NeuroVault. These include statistical maps from the analyses underlying the definition of our regions of interest, as well as the statistical maps from the frequentist analyses presented in Appendix 5. Full fMRI datasets from all participants are accessible at Zenodo.

The following datasets were generated:

| Author(s) | Year | Dataset title | Dataset URL | Database and Identifier |
|---|---|---|---|---|
| Lengersdorff L, Wagner I, Lamm C, Olsson O, Petrovic P | 2019 | Grand Theft Empathy: The effects of violent video games on empathy | https://osf.io/yx423/ | Open Science Framework, yx423 |
| Lengersdorff LL | 2023 | Data from: Grand Theft Empathy? Evidence for the absence of effects of violent video games on empathy for pain and emotional reactivity to violence | https://doi.org/10.5281/zenodo.10057633 | Zenodo, 10.5281/zenodo.10057633 |
| Lengersdorff L | 2023 | Grand Theft Empathy: T-maps | https://neurovault.org/collections/13395/ | NeuroVault, 13395 |

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

# Appendix 1

## Detailed results of the whole-brain analyses for ROI definition

Here, we present the results of the whole-brain analyses underlying our ROI definition (as explained in the main text, section Methods: fMRI analysis: empathy for pain) in more detail. *Appendix 1— table 1* presents the results of the analysis performed on the data from the empathy-for-pain task (only first session). *Appendix 1—table 2* presents the results of the analysis performed on the data from the emotional reactivity task.

**Appendix 1—table 1.** Results of whole-brain analyses for region of interest (ROI) definition, empathy-for-pain task.

Tested contrast: *Other Pain – Other No Pain*, data only taken from the first session. We report the first local maximum within each cluster. Effects were tested for significance with a significance threshold of $p < 0.05$, FWE-corrected. We only report clusters larger than 10 voxels.

| | MNI coordinates | | | | |
|---|---|---|---|---|---|
| Brain region | x | y | z | z-Value | Cluster size |
| R Insula | 30 | 22 | –14 | 6.15 | 27 |
| R Inferior frontal gyrus, orbital part | 50 | 26 | –6 | 6.29 | 125 |
| R Inferior frontal gyrus, opercular part | 50 | 18 | 8 | 5.50 | 23 |
| L Insula | –34 | 20 | 0 | 6.35 | 343 |
| L Superior frontal gyrus, medial part | –2 | 34 | 34 | 6.32 | 86 |
| L Supramarginal gyrus | –58 | –56 | 30 | 6.26 | 110 |
| L Inferior parietal lobule | 58 | –56 | 40 | 5.69 | 12 |

**Appendix 1—table 2.** Results of whole-brain analyses for region of interest (ROI) definition, emotional reactivity task.

Tested contrast: *Violent – Neutral*. We report the first local maximum within each cluster. Effects were tested for significance with a significance threshold of $p < 0.05$, FWE-corrected. We only report clusters larger than 10 voxels.

| | MNI coordinates | | | | |
|---|---|---|---|---|---|
| Brain region | x | y | z | z-Value | Cluster size |
| L/R Thalamus | -6 | –28 | -6 | >10 | 1405 |
| L/R Inferior occipital lobe | –40 | –68 | –10 | >10 | 30,408 |
| R Amygdala | 20 | 0 | –14 | 5.64 | 26 |
| L Amygdala | –20 | –2 | –12 | 6.09 | 13 |
| L/R Hypothalamus | 4 | –4 | –10 | 6.41 | 27 |
| L Precentral gyrus | –44 | 4 | 30 | >10 | 521 |
| R Precentral gyrus | 46 | 8 | 32 | 8.21 | 1229 |
| R Caudate | 14 | 12 | 10 | 5.80 | 33 |
| L Inferior frontal gyrus, triangular part | –42 | 34 | 16 | 7.05 | 159 |
| R Supramarginal gyrus | 66 | –22 | 28 | 6.11 | 87 |
| L Midcingulum | 0 | 4 | 34 | 6.70 | 34 |
| R Supplementary motor area | 6 | 14 | 62 | 6.54 | 116 |
| L Cerebellum | –22 | –38 | –42 | 7.47 | 42 |
| R Cerebellum | 24 | –34 | –42 | 6.97 | 45 |

# Appendix 2

## Detailed results of Bayesian ROI analyses

In the following, we present the results reported in section Results: fMRI data in more detail. *Appendix 2—table 1* presents the results of the analysis performed on the ROI data extracted from the empathy-for-pain task. *Appendix 2—table 2* presents the results of the analysis performed on the ROI data extracted from the emotional reactivity task.

**Appendix 2—table 1.** Posterior parameter estimates and contrasts of models for functional magnetic resonance imaging (fMRI) signal in the empathy-for-pain task.

Dependent variable: fMRI signal extracted from the respective region of interest (ROI) (standardized to unit variance): Fixed effect: terms in standard font describe the mean regression parameter of the respective event averaged across all conditions. Terms in italic font describe the fixed effect of the respective condition on the regression parameter. Factor codings: Group: control game group = –1, violent game group = 1; Session: first session = –1, second session = 1; Intensity: non-painful = –1, painful = 1. $\beta/\sigma_e$: Mean model parameter divided by the mean error standard deviation. Bayes factors were derived from comparing a model where the respective parameter was unrestricted to a model where it was restricted to zero. †These Bayes factors were derived from comparing a model where the parameter was restricted to be negative to a model where it was restricted to zero (one-sided hypothesis test).

| | aMCC | | | Left AI | | | Right AI | | |
|---|---|---|---|---|---|---|---|---|---|
| **Fixed effect** | $\beta/\sigma_e$ | **95% CI** | **BF** | $\beta/\sigma_e$ | **95% CI** | **BF** | $\beta/\sigma_e$ | **95% CI** | **BF** |
| Self cue | 0.25 | (0.09, 0.41) | 34.770 | 0.16 | (0.00, 0.32) | 1.925 | 0.33 | (0.20, 0.45) | >100 |
| *Group* | 0.03 | (–0.13, 0.20) | 0.264 | –0.01 | (–0.17, 0.16) | 0.260 | 0.03 | (–0.10, 0.16) | 0.257 |
| *Intensity* | 0.14 | (0.03, 0.26) | 3.494 | 0.33 | (0.20, 0.45) | >100 | 0.41 | (0.31, 0.51) | >100 |
| *Session* | 0.08 | (–0.03, 0.19) | 0.582 | 0.10 | (–0.01, 0.21) | 1.054 | 0.05 | (–0.06, 0.15) | 0.301 |
| *Group*Intensity* | –0.01 | (–0.13, 0.10) | 0.201 | –0.02 | (–0.14, 0.11) | 0.209 | –0.01 | (–0.11, 0.09) | 0.191 |
| *Group*Session* | 0.05 | (–0.05, 0.16) | 0.276 | 0.00 | (–0.11, 0.10) | 0.179 | –0.03 | (–0.13, 0.07) | 0.236 |
| *Intensity*Session* | –0.02 | (–0.11, 0.08) | 0.182 | –0.08 | (–0.18, 0.02) | 0.577 | –0.01 | (–0.09, 0.07) | 0.150 |
| *Group*Intensity*Session* | –0.03 | (–0.12, 0.07) | 0.193 | –0.04 | (–0.14, 0.06) | 0.253 | –0.03 | (–0.11, 0.05) | 0.241 |
| Other cue | 0.41 | (0.26, 0.56) | >100 | 0.06 | (–0.11, 0.22) | 0.299 | 0.13 | (0.00, 0.26) | 1.459 |
| *Group* | 0.11 | (–0.04, 0.26) | 0.627 | 0.03 | (–0.14, 0.21) | 0.292 | 0.09 | (–0.04, 0.22) | 0.601 |
| *Intensity* | 0.14 | (0.05, 0.24) | 8.814 | 0.19 | (0.10, 0.28) | >100 | 0.20 | (0.12, 0.28) | >100 |
| *Session* | 0.00 | (–0.11, 0.10) | 0.175 | 0.08 | (–0.02, 0.20) | 0.576 | 0.07 | (–0.02, 0.16) | 0.596 |
| *Group*Intensity* | 0.00 | (–0.10, 0.09) | 0.161 | 0.02 | (–0.07, 0.11) | 0.165 | –0.01 | (–0.09, 0.07) | 0.169 |
| *Group*Session* | –0.05 | (–0.16, 0.05) | 0.307 | 0.02 | (–0.08, 0.13) | 0.207 | 0.01 | (–0.08, 0.10) | 0.183 |
| *Intensity*Session* | –0.03 | (–0.13, 0.06) | 0.198 | –0.07 | (–0.17, 0.02) | 0.516 | –0.09 | (–0.17,–0.02) | 2.283 |
| *Group*Intensity*Session* | –0.05 | (–0.14, 0.05) | 0.402† | –0.06 | (–0.15, 0.03) | 0.547† | –0.01 | (–0.09, 0.07) | 0.190† |
| Self stimulation | 1.02 | (0.86, 1.17) | >100 | 1.38 | (1.22, 1.55) | >100 | 0.82 | (0.69, 0.95) | >100 |
| *Group* | –0.16 | (–0.32, 0.00) | 1.690 | 0.01 | (–0.16, 0.18) | 0.286 | –0.02 | (–0.16, 0.12) | 0.245 |
| *Intensity* | 0.53 | (0.39, 0.68) | >100 | 0.63 | (0.49, 0.78) | >100 | 0.64 | (0.51, 0.77) | >100 |
| *Session* | –0.04 | (–0.16, 0.08) | 0.280 | –0.10 | (–0.22, 0.02) | 0.617 | –0.02 | (–0.12, 0.08) | 0.218 |
| *Group*Intensity* | 0.01 | (–0.14, 0.16) | 0.253 | 0.00 | (–0.15, 0.14) | 0.289 | 0.00 | (–0.13, 0.13) | 0.264 |
| *Group*Session* | 0.10 | (–0.02, 0.23) | 0.737 | 0.09 | (–0.03, 0.21) | 0.585 | 0.10 | (0.00, 0.19) | 1.162 |
| *Intensity*Session* | –0.04 | (–0.14, 0.05) | 0.220 | –0.14 | (–0.24,–0.04) | 5.631 | –0.06 | (–0.14, 0.02) | 0.385 |
| *Group*Intensity*Session* | –0.03 | (–0.12, 0.07) | 0.191 | –0.01 | (–0.11, 0.09) | 0.184 | –0.04 | (–0.11, 0.04) | 0.234 |
| Other stimulation | 0.33 | (0.19, 0.48) | >100 | 0.28 | (0.10, 0.46) | 20.821 | 0.23 | (0.07, 0.38) | 12.416 |
| *Group* | 0.13 | (–0.03, 0.28) | 0.852 | 0.09 | (–0.10, 0.27) | 0.490 | 0.07 | (–0.08, 0.23) | 0.378 |
| *Intensity* | 0.32 | (0.23, 0.41) | >100 | 0.39 | (0.30, 0.48) | >100 | 0.36 | (0.27, 0.45) | >100 |
| *Session* | 0.11 | (0.00, 0.22) | 1.236 | 0.07 | (–0.04, 0.18) | 0.455 | 0.05 | (–0.05, 0.14) | 0.286 |

*Appendix 2—table 1 Continued on next page*

Appendix 2—table 1 Continued

| Fixed effect | aMCC $\beta/\sigma_e$ | 95% CI | BF | Left AI $\beta/\sigma_e$ | 95% CI | BF | Right AI $\beta/\sigma_e$ | 95% CI | BF |
|---|---|---|---|---|---|---|---|---|---|
| *Group*Intensity* | 0.05 | (–0.04, 0.14) | 0.290 | –0.03 | (–0.12, 0.05) | 0.191 | 0.02 | (–0.07, 0.12) | 0.231 |
| *Group*Session* | –0.01 | (–0.12, 0.10) | 0.174 | –0.09 | (–0.20, 0.02) | 0.663 | –0.03 | (–0.13, 0.07) | 0.235 |
| *Intensity*Session* | –0.14 | (–0.22,–0.04) | 16.742 | –0.11 | (–0.19,–0.02) | 2.595 | –0.14 | (–0.22,–0.06) | 19.384 |
| *Group*Intensity*Session* | –0.01 | (–0.10, 0.09) | 0.176† | 0.01 | (–0.08, 0.10) | 0.143† | –0.04 | (–0.12, 0.04) | 0.434† |
| Rating | 4.16 | (3.82, 4.48) | >100 | 5.01 | (4.73, 5.28) | >100 | 3.24 | (2.96, 3.52) | >100 |
| *Group* | 0.02 | (–0.17, 0.20) | 0.289 | –0.06 | (–0.25, 0.12) | 0.335 | 0.05 | (–0.09, 0.18) | 0.333 |
| *Session* | –0.16 | (–0.50, 0.18) | 0.700 | 0.10 | (–0.19, 0.37) | 0.475 | –0.06 | (–0.34, 0.22) | 0.464 |
| *Group*Session* | 0.04 | (–0.15, 0.23) | 0.294 | –0.02 | (–0.20, 0.16) | 0.303 | –0.05 | (–0.17, 0.09) | 0.309 |

**Appendix 2—table 2.** Posterior parameter estimates and contrasts of models for functional magnetic resonance imaging (fMRI) signal in the emotional reactivity task.
Dependent variable: fMRI signal extracted from the respective region of interest (ROI) (standardized to unit variance): Fixed effect: terms in standard font describe the mean regression parameter of the respective event averaged across all conditions. Terms in italic font describe the fixed effect of the respective condition on the regression parameter. Factor codings: Group: control game group = –1, violent game group = 1; Content: neutral = –1, violent = 1; Context: real = –1, game = 1. $\beta/\sigma_e$: Mean model parameter divided by the mean error standard deviation. Bayes factors were derived from comparing a model where the respective parameter was unrestricted to a model where it was restricted to zero. †These Bayes factors were derived from comparing a model where the parameter was restricted to be negative to a model where it was restricted to zero (one-sided hypothesis test).

| Fixed effect | Left amygdala $\beta/\sigma_e$ | 95% CI | BF | Right amygdala $\beta/\sigma_e$ | 95% CI | BF |
|---|---|---|---|---|---|---|
| Pictures | 3.59 | (3.21, 3.96) | >100 | 4.25 | (3.91, 4.57) | >100 |
| *Group* | 0.22 | (–0.15, 0.57) | 0.639 | –0.02 | (–0.33, 0.30) | 0.298 |
| *Content* | 1.13 | (0.87, 1.38) | >100 | 1.06 | (0.78, 1.31) | >100 |
| *Context* | –0.01 | (–0.23, 0.22) | 0.252 | –0.06 | (–0.29, 0.17) | 0.281 |
| *Group*Content* | –0.04 | (–0.29, 0.22) | 0.324† | –0.02 | (–0.27, 0.25) | 0.338† |
| *Group*Context* | –0.16 | (–0.38, 0.07) | 0.554 | –0.25 | (–0.48,–0.02) | 2.204 |
| *Content*Context* | –0.06 | (–0.23, 0.12) | 0.227 | 0.00 | (–0.18, 0.18) | 0.205 |
| *Group*Content*Context* | –0.01 | (–0.18, 0.17) | 0.205† | 0.02 | (–0.17, 0.20) | 0.163† |
| Rating | 1.02 | (0.73, 1.30) | >100 | 0.97 | (0.72, 1.24) | >100 |
| *Group* | –0.10 | (–0.36, 0.19) | 0.315 | –0.05 | (–0.30, 0.21) | 0.255 |

# Appendix 3

## Pictures used in the emotional reactivity task

We conducted a pilot study to match pictures taken from the video game to pictures taken from the IAPS (*Lang et al., 2005*). We preselected 33 IAPS pictures of neutral content (people with neutral facial expressions, objects) and 34 IAPS pictures of violent content (dead bodies, mutilations, fights, weapons), as well as 33 game pictures of neutral content and 46 games pictures of violent content. In an online survey, 31 participants (16 female, 15 male) rated these pictures in terms of valence and arousal, using the 9-point self-assessment manikin scale (*Bradley and Lang, 1994*).

We calculated the mean values of valence and arousal across participants per picture, and used these scores, as well as the individual pictures' content, to select 10 pictures per condition as stimuli for the emotional reactivity task. The scores for the final selection of pictures are listed in *Appendix 3—table 1* Note that after matching, there were still systematic differences between game and real pictures: violent real pictures were generally rated higher in arousal, and lower in valence, than violent game pictures. This is due to the fact that a matching purely on valence and arousal scores would have led to sets of pictures with highly different contents (i.e. real pictures showing mostly fights and threats without blood, and game pictures showing mostly dead bodies and highly violent attacks). However, we deemed it important that real pictures and game pictures were also as similar in content as possible. Moreover, we believe that a difference in valence and arousal between real pictures and game pictures is only a minor issue for the experimental design. Our main research question does not concern differences in behavioral and neural responses to real vs. game pictures, but how these responses differ between participants who played a highly violent video game and participants who played a non-violent video game.

The game pictures are available at https://osf.io/yx423/.

**Appendix 3—table 1.** Pictures used in the emotional reactivity paradigm.
Picture ID: International Affective Pictures System (IAPS) ID numbers for pictures of the Neutral Real and Violent Real conditions; arbitrary internal ID numbers for pictures of the Neutral Game and Violent Game conditions.

| Condition | Picture ID | Content | Valence | Arousal |
|---|---|---|---|---|
| Neutral real | 2038 | Woman reading alone | 5.60 | 2.17 |
| | 2200 | Man | 5.52 | 1.71 |
| | 2210 | Man | 5.10 | 2.00 |
| | 2215 | Man | 5.30 | 1.63 |
| | 2383 | Woman on the phone | 5.33 | 1.30 |
| | 2393 | Two workers | 5.37 | 1.37 |
| | 2440 | Woman | 5.48 | 1.42 |
| | 2495 | Man | 5.27 | 1.87 |
| | 2570 | Man | 5.50 | 1.57 |
| | 2749 | Man smoking | 5.33 | 1.90 |
| Violent real | 3010 | Dead body | 1.45 | 6.81 |
| | 3015 | Dead body | 1.32 | 7.16 |
| | 3016 | Dead body | 1.87 | 6.20 |
| | 3060 | Mutilation | 1.42 | 7.00 |
| | 3120 | Mutilation | 1.63 | 6.27 |
| | 6530 | Man hits woman | 2.81 | 4.55 |
| | 6550 | Man threatens woman with knife | 2.06 | 5.7 |
| | 6560 | Man threatens woman with gun | 1.87 | 6.13 |

*Appendix 3—table 1 Continued on next page*

*Appendix 3—table 1 Continued*

| Condition | Picture ID | Content | Valence | Arousal |
|---|---|---|---|---|
| | 6561 | Man hits woman | 3.29 | 4.23 |
| | 6571 | Man threatens man with gun | 3.23 | 4.06 |
| Neutral game | ng1 | Woman | 5.40 | 1.67 |
| | ng2 | Three women smoking | 5.32 | 1.84 |
| | ng3 | Man | 5.13 | 1.68 |
| | ng4 | Woman | 5.40 | 1.80 |
| | ng5 | Woman | 5.39 | 1.35 |
| | ng6 | Man | 5.23 | 1.42 |
| | ng7 | Woman | 5.61 | 1.81 |
| | ng8 | Man | 5.45 | 1.48 |
| | ng9 | Man | 5.65 | 1.48 |
| | ng10 | Two workers | 5.52 | 1.52 |
| Violent game | vg1 | Man attacks man with chainsaw | 2.13 | 5.90 |
| | vg2 | Mutilation | 2.23 | 5.65 |
| | vg3 | Dead body | 2.43 | 5.33 |
| | vg4 | Dead body | 2.52 | 5.16 |
| | vg5 | Dead body | 2.52 | 4.71 |
| | vg6 | Man shoots man in the head | 2.63 | 5.00 |
| | vg7 | Man shoots man in the head | 2.58 | 5.29 |
| | vg8 | Man chokes man | 3.13 | 4.37 |
| | vg9 | Man shoots man in the head | 3.06 | 4.52 |
| | vg10 | Man threatens man with gun | 3.61 | 3.74 |

## Appendix 4

### Bayesian hierarchical models

Behavioral data analysis

We fitted hierarchical censored regression models to the rating data from the empathy-for-pain task and the emotional reactivity task. Ratings were collected using a 100-step VAS, and could thus lie in the range [0,100]. For numerical reasons, we first linearly transformed ratings to the range [–3,3]. In the following, let $i$ index participants, and $t$ index trials. The censored regression model relates $y_{it}$, the rating given by participant $i$ in trial $t$, to a latent response variate $y_{it}^*$ by the function

$$y_{it} = \begin{cases} -3, & y_{it}^* \leq -3 \\ y_{it}^*, & -3 < y_{it}^* < 3 \\ 3, & 3 \leq y_{it}^* \end{cases}$$

For $y_i^*$, the vector of latent responses of participant $i$, we formulate the linear model

$$y_i^* = \mu + X_i\beta + Z_ib_i + \varepsilon_i,$$

where $\mu$ is the grand mean parameter, $X_i$ and $Z_i$ are the th participant's design matrices associated with fixed effects and random effects, respectively, $\beta$ is the vector of fixed effects, $b_i$ is the vector of random effects of participant $i$, and $\varepsilon_i$ is the vector of error terms. Further, we assume

$$\varepsilon_{it} \sim \mathcal{N}\left(0, \sigma_{\varepsilon_i}\right)$$
$$\sigma_{\varepsilon_i} \sim \text{Lognormal}\left(\mu_{\sigma_\varepsilon}, \sigma_{\sigma_\varepsilon}\right),$$

where $\sigma_{\varepsilon_i}$ is the residual error variance associated with participant $i$, and $\mu_{\sigma_\varepsilon}$ and $\sigma_{\sigma_\varepsilon}$ are the hyperparameters of the Lognormal distribution of residual error variances. For these hyperparameters, we formulate the priors

$$\mu_{\sigma_\varepsilon} \sim \mathcal{N}\left(0, \frac{1}{2}\right)$$
$$\sigma_{\sigma_\varepsilon} \sim \text{Gamma}\left(4 \cdot \log\left(2\right), 4\right),$$

which put the majority of their mass on sensible values. For the vector of random effects, we assume

$$b_i \sim \mathcal{N}_{k_b}\left(0, D \cdot R \cdot D\right),$$
$$D = \text{diag}\left(\sigma_{b_1}, ..., \sigma_{b_{k_b}}\right)$$

where $k_b$ is the number of random effects, $D$ is the diagonal matrix with the random effect standard deviations $\sigma_{b_1}, ..., \sigma_{b_{k_b}}$ on the diagonal, and $R$ is the correlation matrix of random effects. We further formulate the weakly informative priors

$$\sigma_{b_1}, ..., \sigma_{b_{k_b}} \sim \text{Halfnormal}\left(0, 1\right),$$
$$R \sim \text{LKJ}\left(2\right).$$

Lastly, we formulate the following prior on the fixed effects,

$$\mu \sim \mathcal{N}\left(0, 3\right),$$
$$\frac{\beta}{\bar{\sigma}_\varepsilon} \sim \mathcal{N}_{k_\beta}\left(0, \frac{1}{2} \cdot I_{k_\beta}\right),$$
$$\bar{\sigma}_\varepsilon = \exp\left(\mu_{\sigma_\varepsilon} + \frac{\sigma_{\sigma_\varepsilon}^2}{2}\right)$$

where $k_\beta$ is the number of fixed effects, and $\bar{\sigma}_\varepsilon$ is the theoretical mean of error standard deviations across participants. Putting the prior on the ratio $\frac{\beta}{\bar{\sigma}_\varepsilon}$ instead of $\beta$ allows us to formulate an appropriately informed prior without prior knowledge of the average variance of the error term. We use the scaling factor 1/2 to represent our prior assumption that fixed effects are unlikely to be much larger (in absolute value) than the average error standard deviation.

## fMRI data analysis

We fitted hierarchical regression models to the BOLD response data extracted from the ROIs. To account for the autocorrelation that is to be expected in fMRI data, we assumed that the residual error terms within a run could be described by an autoregressive process of order 1 (AR(1)).

To facilitate interpretation and formulation of priors, we directly parameterized the model in terms of contrasts of regression weights. For each subject $i$ and session $j$, we first build the raw design matrix $X_{ij}^R$ as is done in established software such as SPM (Wellcome Trust Centre for Neuroimaging, https://www.fil.ion.ucl.ac.uk/spm). Shortly, for each separate type of event, we created a regressor representing the expected BOLD signal induced by the event by convolving a boxcar function of appropriate onset and length with the canonical hemodynamic response function. Then we constructed the design matrix in terms of contrasts $X_{ij}^C$ by right-multiplying $X_{ij}^R$ with a matrix $C$ that encoded the contrasts of interest. For example, consider the raw design matrix $X_{ij}^R$ of the empathy-for-pain paradigm with the following mapping between column numbers and events:

$$
\begin{aligned}
1 &\mapsto CueSelf: NoPain \\
2 &\mapsto CueSelf: Pain \\
3 &\mapsto CueOther: NoPain \\
4 &\mapsto CueOther: Pain \\
5 &\mapsto StimulusSelf: NoPain \\
6 &\mapsto StimulusSelf: Pain \\
7 &\mapsto StimulusOther: NoPain \\
8 &\mapsto StimulusOther: Pain \\
9 &\mapsto Rating.
\end{aligned}
$$

Then

$$
X_{ij}^R C = X_{ij}^R
\begin{pmatrix}
1 & -1 & & & & & & & \\
1 & 1 & & & & & & & \\
& & 1 & -1 & & & & & \\
& & 1 & 1 & & & & & \\
& & & & 1 & -1 & & & \\
& & & & 1 & 1 & & & \\
& & & & & & 1 & -1 & \\
& & & & & & 1 & 1 & \\
& & & & & & & & 1
\end{pmatrix}
$$

results in a matrix $X_{ij}^C$ with the column-to-contrast mapping

$$
\begin{aligned}
1 &\mapsto CueSelf: Mean \\
2 &\mapsto CueSelf: Pain - NoPain \\
3 &\mapsto CueOther: Mean \\
4 &\mapsto CueOther: Pain - NoPain \\
5 &\mapsto StimulusSelf: Mean \\
6 &\mapsto StimulusSelf: Pain - NoPain \\
7 &\mapsto StimulusOther: Mean \\
8 &\mapsto StimulusOther: Pain - NoPain \\
9 &\mapsto Rating.
\end{aligned}
$$

Finally, for tasks with two sessions (i.e. the empathy-for-pain task), main effects and interactions with the session factor were represented by constructing the final first-level design matrix as the block matrix

$$\mathbf{X}_i = \begin{pmatrix} X_{i1} \\ X_{i2} \end{pmatrix} = \begin{pmatrix} X_{i1}^C & -X_{i1}^C \\ X_{i2}^C & X_{i2}^C \end{pmatrix}.$$

With this construction, the first half of columns of $X_i$ correspond to the effects of events marginal to the session factor, while the second half of columns correspond to the interactions between events and the session factor.

For describing the hierarchical model, we denote the sequence of extracted signals of participant $i$ in session $j$ as $y_{ij}$. Further, we let $t$ index the timepoints within each such sequence, such that $y_{ij} = \left(y_{ij1}, y_{ij2}, ..., y_{ijt}, ...\right)'$. To account for low-frequency changes of signal of no interest (e.g. scanner drift), each $y_{ij}$ was first filtered with a high-pass filter of period 128. As differences in grand mean between participants and sessions were not of interest, each sequence $y_{ij}$ was mean centered to have mean 0. Further, to change the arbitrary scale of BOLD response signals to a known scale, each $y_{ij}$ was scaled to have variance 1. The same operations were also performed on each design matrix $X_{ij}$.

For each $y_{ij}$, we formulate the linear model

$$y_{ij} = X_{ij} \cdot \left(\beta + G_i\gamma + b_i\right) + u_{ij},$$

where $\beta$ is the vector of fixed effects of contrasts, $G_i$ is a variable that takes the value −1 when participant $i$ is in the control group, and 1 when participant $i$ is in the experimental group, $\gamma$ is the vector of fixed effects of the factor group, $b_i$ is the vector of random effects of participant $i$, and $u_{ij}$ is the vector of error terms of participant $i$ in session $j$. We assume that $u_{ij}$ follows an AR(1) process, thus

$$u_{ijt} = \varepsilon_{ijt} + \varphi_{ij}\varepsilon_{ij(t-1)},$$
$$\varepsilon_{ijt} \sim \mathcal{N}\left(0, \sigma_{\varepsilon ij}\right),$$

where $\varphi_{ij}$ is the autoregression parameter of person $i$ in session $j$, $\varepsilon_{ijt}$ are independently and identically distributed impulses, and $\sigma_{\varepsilon ij}$ is the standard deviation of these impulses for participant $i$ and session $j$. Following the hierarchical modeling approach, we assume that $\varphi_{ij}$ and $\sigma_{\varepsilon ij}$ are themselves drawn from a higher-order distribution, for whose hyperparameters we formulate weak priors:

$$\frac{\varphi_{ij} + 1}{2} \sim \text{Logitnormal}\left(\mu_\varphi, \sigma_\varphi\right),$$
$$\mu_\varphi \sim \mathcal{N}\left(0, 1\right),$$
$$\sigma_\varphi \sim \text{Halfnormal}\left(0, 1\right),$$

and

$$\sigma_{\varepsilon ij} \sim \text{Lognormal}\left(\mu_{\sigma_\varepsilon}, \sigma_{\sigma_\varepsilon}\right),$$
$$\mu_{\sigma_\varepsilon} \sim \mathcal{N}\left(-1/2, 1/2\right),$$
$$\sigma_{\sigma_\varepsilon} \sim \text{Gamma}\left(4 \cdot \log\left(2\right), 4\right).$$

For the vector of random effects, we assume

$$b_i \sim \mathcal{N}_{k_b}\left(0, D \cdot R \cdot D\right),$$
$$D = \text{diag}\left(\sigma_{b_1}, ..., \sigma_{b_{k_b}}\right)$$

where $k_b$ is the number of random effects, $D$ is the diagonal matrix with the random effect standard deviations $\sigma_{b_1}, ..., \sigma_{b_{k_b}}$ on the diagonal, and $R$ is the correlation matrix of random effects. We further formulate the priors

$$R \sim \mathrm{LKJ}\left(2\right),$$
$$\sigma_{b_1}, ..., \sigma_{b_{kb}} \sim \mathrm{Halfnormal}\left(0, \frac{\bar{\sigma}_\varepsilon}{2}\right),$$
$$\bar{\sigma}_\varepsilon = \exp\left(\mu_{\sigma_\varepsilon} + \frac{\sigma_{\sigma_\varepsilon}^2}{2}\right),$$

where $\bar{\sigma}_\varepsilon$ is the theoretical mean of error standard deviations across participants. This reflects the assumption that random effects will not be much larger than the mean error standard deviation. Lastly, we formulate the following priors on the fixed effects,

$$\frac{\beta}{\bar{\sigma}_\varepsilon} \sim \mathcal{N}_{k_\beta}\left(0, \frac{1}{10} \cdot I_{k_\beta}\right),$$
$$\frac{\gamma}{\bar{\sigma}_\varepsilon} \sim \mathcal{N}_{k_\gamma}\left(0, \frac{1}{10} \cdot I_{k_\gamma}\right),$$

where $k_\beta$ and $k_\gamma$ are the number of fixed effects. Putting the prior on the ratio $\frac{\beta}{\bar{\sigma}_\varepsilon}$ instead of $\beta$ allows us to formulate an appropriately informed prior without prior knowledge of the average variance of the error term. We use the scaling factor 1/10 to represent our prior assumption that fixed effects are likely to be much smaller (in absolute value) than the average error standard deviation, due to the high amount of noise in fMRI signal. However, as this prior formulation might still have been too vague for proper hypothesis testing via the BF, we additionally informed the prior with a fraction of 2/$n$ of the likelihood of the data (where $n$ is the sample size), therefore calculating fractional BFs (*O'Hagan, 1995*).

## Appendix 5

### Frequentist analyses: behavioral data

As equivalent frequentist analyses of our behavioral data, we fitted linear mixed effects models to the collected ratings. All models were estimated using the R package *lmerTest* (**Kuznetsova et al., 2017**). p-Values were derived using the Satterthwaite approximation of degrees of freedom. We used the conventional significance level of $\alpha$=0.05, and one-sided testing for directional hypotheses.

### Empathy for pain

To analyze the painfulness and unpleasantness ratings collected during the empathy-for-pain paradigm, we modeled fixed effects for the experimental factors *Group* (non-violent vs. violent gaming, coded as –1 and 1), *Time* (pre vs. post gaming sessions, coded as –1 and 1), and *Intensity* (non-painful vs. painful stimulation of the confederate, coded as –1 and 1), as well as all interactions between these factors. Additionally, we modeled per-subject random effects of *Time*, *Intensity*, and these factors' interaction term. *Appendix 5—table 1* presents the results of these analyses. For both ratings, we observed a non-significant *Group\*Session\*Intensity* interaction (for painfulness: one-sided p-value = 0.080; for unpleasantness: one-sided p-value = 0.381), implying no evidence for a VVG effect on behavioral correlates of empathy for pain.

#### Emotional reactivity

To analyze the unpleasantness ratings collected during the emotional reactivity paradigm, we modeled fixed effects for the experimental factors *Group* (non-violent vs. violent gaming, coded as –1 and 1), *Content* (neutral vs. violent, coded as –1 and 1), and *Context* (real vs. game, coded as –1 and 1). Additionally, we modeled per-subject random effects for *Content*, *Context*, and their interaction. *Appendix 5—table 2* presents the results of these analyses. We observed no significant Group\*Content interaction (one-sided p-value = 0.163) or Group\*Content\*Context interaction (one-sided p-value = 0.481), implying no evidence for a VVG effect on behavioral correlates on emotional responses to violent images.

### Frequentist analyses: fMRI data

We performed GLM-based whole-brain analysis using SPM12 (Wellcome Trust Centre for Neuroimaging, https://www.fil.ion.ucl.ac.uk/spm), implemented in MATLAB 2017b. Parameter estimates were estimated on the first level, and the contrasts of interest were then subjected to two-sample *t*-tests on the second level. To increase power to detect effects in our a priori defined ROIs, we used small-volume correction, using the same ROIs as for the Bayesian analyses in the main text. We tested for voxels that survived family-wise error correction, p<0.05. To give a more complete picture, we also tested for voxels inside the ROIs that survived the more lenient thresholds of p<0.001 uncorrected and p<0.05 uncorrected.

#### Empathy for pain

For each participant and session, the first-level design matrix included regressors for the Cue and Stimulation events, separate for all four combinations of conditions (Self No Pain; Self Pain; Other No Pain; Other Pain). As nuisance regressors, we included regressors for the rating events. We then subjected the beta images of the first-level contrast [*Other Pain – Other No Pain*]$_{Session\ 2}$ – [*Other Pain – Other No Pain*]$_{Session\ 1}$ to a two-sample *t*-test, and identified the voxels in which the contrast *Control Group>Experimental Group* was significant and positive.

Using family-wise error correction, we found no significant clusters in any of the three ROIs (aMCC, left AI, right AI). There were also no voxels surviving the uncorrected threshold of p<0.001. In left AI, one voxel out of 343 survived the uncorrected threshold of p<0.05.

Other whole-brain results may be investigated using the T-map provided online (https://neurovault.org/collections/13395/).

#### Emotional reactivity

For each participant, the design matrix included regressors for the blocks of picture presentations, separate for all four combinations of conditions (Neutral Real, Neutral Game, Violent Real, Violent Game). As nuisance regressors, we included regressors for the rating events. We then subjected the

beta images of the first-level contrasts of interest to a two-sample *t*-test, and identified the voxels in which the contrast *Control Group>Experimental Group* was significant and positive.

For the interaction *Group\*Content* (testing whether participants in the violent game group had decreased responses to violent images in general), the contrast of interest was [*Violent Real + Violent Game*]/2 – [*Neutral Real +Neutral Game*]/2. Using family-wise error correction, we found no significant clusters in any of the two ROIs (left amygdala, right amygdala). There were also no voxels surviving the uncorrected threshold of p<0.001. In the left amygdala, 3 voxels out of 220 survived the uncorrected threshold of p<0.05, and in the right amygdala, 10 voxels out of 248 survived this more lenient threshold.

For the interaction *Group\*Content\*Context* (testing whether participants in the violent game group had decreased responses to specifically violent game images), the contrast of interest was [*Violent Game – Neutral Game*] – [*Violent Real – Neutral Real*]. Using family-wise error correction, we found no significant clusters in any of the two ROIs (left amygdala, right amygdala). There were also no voxels surviving the uncorrected threshold of p<0.001. In the left amygdala, 15 voxels out of 220 survived the uncorrected threshold of p<0.05, and in the right amygdala, 49 voxels out of 248 survived this more lenient threshold.

Other whole-brain results may be investigated using the T-maps provided online (https://neurovault.org/collections/13395/).

**Appendix 5—table 1.** Linear mixed effects models for ratings in the empathy-for-pain task.
Dependent variable: empathy ratings (visual analog scale, range: 0–100). Factor codings: Group: control game group = –1, violent game group = 1; Session: first session = –1, second session = 1; Intensity: non-painful = –1, painful = 1.

| Fixed effect | $\beta$ | SE | df | t | p |
|---|---|---|---|---|---|
| Painfulness ratings | | | | | |
| Group | 0.03 | 0.74 | 87.7 | 0.04 | 0.967 |
| Session | 0.16 | 0.43 | 85.9 | 0.38 | 0.706 |
| Intensity | 25.49 | 0.98 | 85.7 | 25.93 | <0.001 |
| Group\*Session | –0.68 | 0.43 | 85.9 | –1.56 | 0.122 |
| Group\*Intensity | –0.92 | 0.98 | 85.7 | –0.96 | 0.340 |
| Session\*Intensity | –0.23 | 0.47 | 82.0 | –0.50 | 0.621 |
| Group\*Session\*Intensity | –0.67 | 0.47 | 82.0 | –1.42 | 0.160 |
| Unpleasantness ratings | | | | | |
| Group | –1.06 | 1.46 | 86.36 | –0.73 | 0.469 |
| Session | –0.81 | 0.55 | 82.36 | –1.47 | 0.145 |
| Intensity | 14.97 | 1.11 | 86.68 | 13.51 | <0.001 |
| Group\*Session | –1.04 | 0.55 | 82.36 | –1.89 | 0.063 |
| Group\*Intensity | –0.43 | 1.11 | 86.68 | –0.39 | 0.696 |
| Session\*Intensity | –0.99 | 0.43 | 84.78 | –2.31 | 0.023 |
| Group\*Session\*Intensity | –0.13 | 0.43 | 84.78 | –0.30 | 0.762 |

**Appendix 5—table 2.** Linear mixed effects models for unpleasantness ratings in the emotional reactivity task.
Dependent variable: unpleasantness ratings (visual analog scale, range: 0–100). Factor codings: Group: control game group = –1, violent game group = 1; Content: neutral = –1, violent = 1; Context: real = –1, game = 1.

| Fixed effect | $\beta$ | SE | df | t | p |
|---|---|---|---|---|---|
| Group | 0.96 | 1.48 | 87 | 0.65 | 0.519 |

*Appendix 5—table 2 Continued on next page*

*Appendix 5—table 2 Continued*

| Fixed effect | β | SE | df | t | p |
|---|---|---|---|---|---|
| *Content* | 29.17 | 1.50 | 87 | 19.40 | <0.001 |
| *Context* | −5.03 | 0.71 | 87 | −7.10 | <0.001 |
| *Group*Content* | 1.49 | 1.50 | 87 | 0.99 | 0.325 |
| *Group*Context* | −0.58 | 0.71 | 87 | −0.82 | 0.415 |
| *Content*Context* | −4.80 | 0.67 | 87 | −7.18 | <0.001 |
| *Group*Content*Context* | −0.03 | 0.67 | 87 | −0.05 | 0.962 |

## Appendix 6

### Covariate analyses

As described in our registration, we additionally performed analyses to investigate the role of trait neuroticism and executive control on the possible VVG effect on empathy. As a measure of trait neuroticism, we used the Neuroticism scale of the German version of the NEO-FFI (*Borkenau and Ostendorf, 1993*). Due to technical issues, the neuroticism measure could not be obtained from seven participants, leaving a sample size of *N*=82 for these analyses. As a measure of executive control, we used stop-signal reaction time (SSRT), measured with the software STOP-IT (*Verbruggen et al., 2008*). Due to technical issues, valid SSRT measures of eight participants were missing. Additionally, we removed SSRT measures of three participants who inhibited responses in significantly more or less than 50% of times (see *Verbruggen et al., 2008*, for an explanation of this criterion). In total, the sample size for analyses involving SSRT was *N*=78.

We added the fixed effects of Neuroticism/SSRT, as well as its interactions with other factors, to the models described in section Methods: Behavioral data analysis. Results are shown in *Appendix 6—table 1* for covariate Neuroticism, and in *Appendix 6—table 2* for covariate SSRT. The BF of the tests of the interaction *Neuroticism\*Group\*Intensity\*Session* was 0.265 for painfulness ratings, and 0.466 for unpleasantness ratings. The BF of the tests of the interaction *SSRT\*Group\*Intensity\*Session* was 0.128 for painfulness ratings, and 0.021 for unpleasantness ratings. This indicates that behavioral VVG effects could also not be observed in participants with high levels of trait neuroticism resp. high levels of SSRT.

To analyze neural responses, we added the fixed effects of Neuroticism/SSRT, as well as its interactions with other factors, to the models described in section Methods: fMRI analyses: empathy for pain. For *Other Cue* events, the BFs of the tests of the interaction *Neuroticism\*Group\*Intensity\*Session* were $BF_{aMCC} = 1.517$; $BF_{left\ AI} = 0.901$; $BF_{right\ AI} = 0.703$. For *Other Stimulation* events, the BFs of the tests of the same interaction were $BF_{aMCC} = 0.348$; $BF_{left\ AI} = 0.226$; $BF_{right\ AI} = 0.209$. Thus, our data give mixed levels of evidence for the absence of a modulation of VVG effects through trait neuroticism. For *Other Cue* events, the data is inconclusive: we cannot confidently say that there is indeed no modulation of the VVG effect on brain activity during cues that indicate whether or not the other person will receive a painful stimulus. For *Other Stimulation* events, we obtain moderate evidence for the absence of such a modulation: we can, with some confidence, say that participants with high neuroticism were not more susceptible to VVG effects on brain activity while the other person received painful stimulation.

For *Other Cue* events, the BFs of the tests of the interaction *SSRT\*Group\*Intensity\*Session* were $BF_{aMCC} = 0.087$; $BF_{left\ AI} = 0.126$; $BF_{right\ AI} = 0.329$. For *Other Stimulation* events, the BFs of the tests of the same interaction were $BF_{aMCC} = 0.936$; $BF_{left\ AI} = 0.316$; $BF_{right\ AI} = 0.551$. Thus, our data give mixed levels of evidence for the absence of a modulation of VVG effects through executive control.

For *Other Cue* events, we obtain moderate to substantial evidence for the absence of such a modulation: we can, with some confidence, say that participants with low executive control were not more susceptible to VVG effects on brain activity during cues that indicate whether or not the other person will receive a painful stimulus. For *Other Cue* events, the data is inconclusive: we cannot confidently say that there is indeed no modulation of the VVG effect on brain activity while the other person received painful stimulation.

**Appendix 6—table 1.** Posterior parameter means of models for ratings in the empathy-for-pain task, including the trait covariate Neuroticism.

Dependent variable: empathy ratings (visual analog scale, range: 0–100). Factor codings: Group: control game group = –1, violent game group = 1; Session: first session = –1, second session = 1; Intensity: non-painful = –1, painful = 1. The variable Neuroticism was scaled to mean 0 and standard deviation 1. Bayes factors were derived from comparing a model where the respective parameter was unrestricted to a model where it was restricted to zero. †These Bayes factors were derived from comparing a model where the parameter was restricted to be negative to a model where it was

restricted to zero (one-sided hypothesis test).

| Fixed effect | β | 95% Credible interval | | Bayes factor |
| --- | --- | --- | --- | --- |
| **Painfulness ratings** | | | | |
| Group | 0.622 | −1.147 | 2.385 | 0.138 |
| Neuroticism | −0.624 | −2.401 | 1.135 | 0.140 |
| Session | 0.518 | −0.240 | 1.270 | 0.105 |
| Intensity | 27.208 | 24.593 | 29.772 | >100 |
| Group*Neuroticism | −1.767 | −3.597 | 0.024 | 0.541 |
| Group*Session | −0.506 | −1.278 | 0.239 | 0.100 |
| Neuroticism*Session | −0.691 | −1.411 | 0.009 | 0.225 |
| Group*Intensity | −1.253 | −3.788 | 1.276 | 0.255 |
| Neuroticism*Intensity | 0.537 | −2.022 | 3.069 | 0.174 |
| Session*Intensity | −0.374 | −1.521 | 0.791 | 0.082 |
| Group*Neuroticism*Session | −0.155 | −0.898 | 0.578 | 0.046 |
| Group*Neuroticism*Intensity | −0.383 | −2.999 | 2.205 | 0.158 |
| Group*Intensity*Session | −0.599 | −1.743 | 0.506 | 0.112 |
| Neuroticism*Intensity*Session | 0.628 | −0.536 | 1.826 | 0.113 |
| Group*Neuroticism*Intensity*Session | −0.717 | −1.865 | 0.473 | 0.265† |
| **Unpleasantness ratings** | | | | |
| Group | −0.562 | −4.120 | 3.058 | 0.202 |
| Neuroticism | 3.430 | −0.141 | 7.296 | 1.007 |
| Session | −0.258 | −1.534 | 1.131 | 0.081 |
| Intensity | 17.004 | 14.246 | 19.772 | >100 |
| Group*Neuroticism | −2.029 | −5.759 | 1.932 | 0.390 |
| Group*Session | −0.734 | −2.060 | 0.589 | 0.127 |
| Neuroticism*Session | −1.682 | −3.021 | −0.377 | 1.697 |
| Group*Intensity | −0.875 | −3.536 | 1.770 | 0.181 |
| Neuroticism*Intensity | 1.977 | −0.917 | 4.697 | 0.390 |
| Session*Intensity | −1.192 | −2.124 | −0.269 | 1.236 |
| Group*Neuroticism*Session | −0.904 | −2.301 | 0.443 | 0.168 |
| Group*Neuroticism*Intensity | 0.496 | −2.258 | 3.253 | 0.167 |
| Group*Intensity*Session | −0.246 | −1.161 | 0.696 | 0.060 |
| Neuroticism*Intensity*Session | −0.464 | −1.402 | 0.486 | 0.081 |
| Group*Neuroticism*Intensity*Session | −0.850 | −1.796 | 0.092 | 0.466† |

**Appendix 6—table 2.** Posterior parameter means of models for ratings in the empathy-for-pain task, including the trait covariate SSRT (stop-signal reaction time).
Dependent variable: empathy ratings (visual analog scale, range: 0–100). Factor codings: Group: control game group = −1, violent game group = 1; Session: first session = −1, second session = 1; Intensity: non-painful = −1, painful = 1. The variable SSRT was scaled to mean 0 and standard deviation 1. Bayes factors were derived from comparing a model where the respective parameter was unrestricted to a model where it was restricted to zero. †These Bayes factors were derived from comparing a model where the parameter was restricted to be negative to a model where it was restricted to zero (one-sided hypothesis test).

| Fixed effect | $\beta$ | 95% Credible interval | | Bayes factor |
|---|---|---|---|---|
| **Painfulness ratings** | | | | |
| Group | 0.990 | −1.016 | 3.014 | 0.173 |
| SSRT | 0.060 | −2.183 | 2.346 | 0.128 |
| Session | 0.529 | −0.374 | 1.431 | 0.101 |
| Intensity | 27.712 | 25.107 | 30.359 | >100 |
| Group*SSRT | −0.413 | −2.724 | 1.953 | 0.139 |
| Group*Session | −0.539 | −1.517 | 0.363 | 0.097 |
| SSRT*Session | −0.290 | −1.389 | 0.794 | 0.068 |
| Group*Intensity | −1.590 | −4.051 | 0.933 | 0.294 |
| SSRT*Intensity | 0.291 | −2.671 | 3.222 | 0.165 |
| Session*Intensity | 0.051 | −1.035 | 1.145 | 0.063 |
| Group*SSRT*Session | 0.587 | −0.440 | 1.611 | 0.105 |
| Group*SSRT*Intensity | 2.030 | −0.898 | 4.986 | 0.402 |
| Group*Intensity*Session | −0.830 | −1.944 | 0.230 | 0.173 |
| SSRT*Intensity*Session | 1.138 | −0.170 | 2.400 | 0.333 |
| Group*SSRT*Intensity*Session | −0.409 | −1.710 | 0.861 | 0.128† |
| **Unpleasantness ratings** | | | | |
| Group | −0.080 | −4.225 | 3.970 | 0.214 |
| SSRT | 0.661 | −3.864 | 5.040 | 0.262 |
| Session | −0.476 | −1.878 | 1.018 | 0.103 |
| Intensity | 17.149 | 14.265 | 19.969 | >100 |
| Group*SSRT | −3.128 | −7.644 | 1.276 | 0.613 |
| Group*Session | −0.817 | −2.328 | 0.692 | 0.151 |
| SSRT*Session | 0.688 | −0.962 | 2.386 | 0.127 |
| Group*Intensity | −1.531 | −4.313 | 1.325 | 0.259 |
| SSRT*Intensity | 1.193 | −1.907 | 4.284 | 0.216 |
| Session*Intensity | −1.089 | −2.014 | −0.211 | 0.822 |
| Group*SSRT*Session | 1.253 | −0.465 | 2.938 | 0.269 |
| Group*SSRT*Intensity | 1.527 | −1.569 | 4.545 | 0.277 |
| Group*Intensity*Session | −0.092 | −1.015 | 0.834 | 0.051 |
| SSRT*Intensity*Session | −0.137 | −1.136 | 0.900 | 0.055 |
| Group*SSRT*Intensity*Session | 0.999 | −0.040 | 2.031 | 0.021† |

