## [Editor Report]

Lengersdorff and colleagues present behavioural and fMRI data that are valuable in demonstrating no impact of violent video games on the emotional response to pain in their particular sample. The effects may be specific to the participant group who have no neurological disorder and no character traits that would predispose to desensitisation (because they are selected due to little prior experience playing these games), and there are some openly-discussed test-retest reliability issues (session 1->2) with the fMRI measures, but they present convincing evidence for the absence of effect in this group.

---

## [Decision Letter]

**Decision letter after peer review:**

Thank you for submitting your article "Grand Theft Empathy? Evidence for the absence of effects of violent video games on empathy for pain and emotional reactivity to violence" for consideration by *eLife*. Your article has been reviewed by 3 peer reviewers, one of whom is a member of our Board of Reviewing Editors, and the evaluation has been overseen by Jonathan Roiser as the Senior Editor.

Essential revisions:

1. Abstract. Given the potentially damaging nature of conclusions that there is no impact of VGGs on emotional response in general, we would like it to be mentioned in the abstract the groups/settings to which this may not generalise.

2. Generalisation. The authors already discuss that the findings may not generalise to clinical groups, younger groups, or longer/different types of gameplay. However, all reviewers felt that greater emphasis should be placed on this issue, especially due to the danger associated with misinterpretation of the present dataset. They would therefore like to see greater discussion of:

(a) The authors picked 18-35 year old males who had not played VVGs at least 12 months before testing, nor played Grand Theft Auto ever before. This is one of the most popular games of all time and it presumably highly constrained their sample. We understand why they did this with respect to control, but wonder whether they also therefore preselect for a certain phenotype who is less likely to desensitise. We feel that the authors should discuss that the findings may not generalise to those who would actually choose to play these games (see below for details).

(b) They discuss that effects may differ with more video game exposure, but it may be worth properly considering that the level of exposure in the present study is arguably lower than among a large number of gamers – for whom the impact of VVGs is of greatest concern. It would also improve the manuscript to include discussion of how the quantity of exposure was decided. It is not clear if this decision was governed by feasibility constraints alone, or is there any prior literature that points to this dosage.

(c) The game played by the participants lacked many of the core features of popular video games such as immersion in a complex context/story, emotional investment and role-playing, progression and conflict between objectives, an internal motivation to play, etc.

(d) The link between laboratory measures of empathy and emotional reactivity and real-world behaviour is not supported by strong evidence and as such it is unclear what changes (or no changes) on these measures would imply for the impact of violent video games on psychological functioning.

3. Mechanism. The authors consider how it is important to dissociate persistent effects of VGGs from short term effects such as arousal or priming. This could do with some greater clarification for the reader. Specifically, while we must dissociate effects half an hour later from those that persist for longer, one could easily describe longer-term impacts that operate due to arousal or priming mechanisms. In clarifying the argument, it would be helpful to know whether these previous experimental studies have found influences, or no influences, of the gaming. Further details are provided below.

4. Analysis specifics. More details about the parameters in the Bayesian tests and power calculations are requested. Additionally, despite the authors' truly commendable sample size, it must be noted that this is still insufficiently powered to detect small effects. And might a small effect with such implications still be important?

5. Test. The impact of playing violent (/non-violent) video games is measured using the empathy for pain paradigm in this study. It would be useful to note the test-retest reliability of the neural response in the empathy for pain paradigm. Relatedly, is there any data to suggest that this neural response is sensitive to change in any external factors? Furthermore, the use of two paradigms to measure the response to video games allows one to test the generalisability of the results. It would be useful to note if there is any relationship between the key metrics from these two paradigms, i.e. do individuals who show a greater empathy for pain (neurally/behaviourally) also show greater emotional reactivity?

6. We have also outlined some smaller changes that we believe would improve the manuscript, which are detailed in the comments below.

*Reviewer #1 (Recommendations for the authors):*

The authors provide compelling support for the idea that playing VGGs yields no influence on one's ratings of painfulness or unpleasantness when observing others in pain, or unpleasantness ratings when observing videos of violence – in their particular sample of interest. fMRI data showed no difference in regions responsive to others' pain, or those responsive to observed violence, as a function of VGG.

I was very impressed with this manuscript. It was a joy to read due to its clarity of expression and simple paradigm answering a clear theoretical question. I found the data compelling, in an impressively large sample, and believe the findings will inform this thorny debate and be of interest to a wide readership. I was especially impressed with the crystal clear manner in which the authors highlighted previous work, along with the questions that remain unanswered and how this study will answer them. Such a rewarding read, and notably unusual for authors to make the life of the reader so easy in this manner. Thank you.

I just have a couple of suggestions for improvement, relating to the underlying mechanism and generalisability.

Mechanism. The authors consider how it is important to dissociate persistent effects of VGGs from short term effects such as arousal or priming. I think I have some idea what the authors mean but feel it could do with some greater clarification for the reader. Specifically, of course, we must dissociate effects half an hour later from those that persist for longer, and I understand the main point. However, one could easily describe longer-term impacts that operate due to arousal or priming mechanisms. I think it is important for the reader to outline the specific mechanistic account that would have short-term, but not long-term, impact. Presumably, we are talking some type of habituation/adaptation mechanism, whereby activation of structures such as the amygdala during gameplay means that they cannot be activated subsequently so readily? I don't quite follow this, especially given that presumably there is evidence of activation of these structures to observation of pain but it's the mediation by the gaming that differs. In clarifying the argument, it would be helpful to know whether these previous experimental studies have found influences, or no influences, of the gaming.

Generalisation. The authors already discuss that the findings may not generalise to clinical groups, younger groups, or longer/different types of gameplay. Another dimension jumped out to me and I think the manuscript would benefit from its discussion. Specifically, they pick 18-35-year-old males who had not played VVGs at least 12 months before testing, and not ever played Grand Theft Auto. I understand why they did this with respect to control, but I wonder whether they also, therefore, preselect for a certain phenotype who is less likely to desensitise. If they have not chosen to play these types of game this perhaps means they do not receive the positive emotional reward of violence that some other participants may do. I imagine that for large swathes of the population the negative emotional response to observing death is so extreme that they would obtain no reward whatsoever from these games. These are possibly the individuals tested here, and their response could prove less relevant to understanding those who actually play them – and for whom this debate has practical implications. Those who receive the intrinsic positive reward, due to feelings of power or the like, may desensitise in a way that this sample do not. Therefore, while I think these findings are highly compelling in this sample, I think the authors should discuss that the findings may not generalise to those who would actually choose to play these games.

Abstract. Given the potentially damaging nature of conclusions that there is no impact of VGGs on emotional response in general, I think it is wise to mention in the abstract the groups/settings to which this may not generalise.

I note in Figure 2 that all trends are in predicted directions, although I guess the Bayesian stats suggest this is irrelevant. Worth a comment?

*Reviewer #2 (Recommendations for the authors):*

Lengersdorff and colleagues conducted a study to examine the impact of exposure to video-game violence on physiological (fMRI) and behavioral measures of empathic responses and emotional reactivity. In the experimental study, the authors initially assessed empathic responses and emotional reactivity in 89 participants at baseline. Then, roughly half of the participants were randomly assigned to either a violent video game condition or a control condition. During the intervention phase, participants in the violent condition played a modified video game that incentivized graphical acts of interpersonal violence for seven hours over two weeks. Meanwhile, participants in the control condition played a similar game but were asked to take pictures instead of committing violence. The findings of the study suggest that this intervention had no effect on brain responses or behavioral responses previously linked to empathy for pain in others, or on responses linked to emotional reactivity.

The data from this study offer valuable insights into the impact of brief exposure to video game violence on laboratory measures of empathy and emotional reactivity and the results appear to be free of confounds. However, it should be noted that the study's ability to shed light on the broader impact of video games on psychological functioning outside of the laboratory is limited.

Strengths:

1. The authors were able to enhance the internal validity of their research by developing an experiment that included a well-crafted control condition, pre-post measures, and a carefully chosen participant population. This approach enabled them to minimize the influence of many confounding factors that frequently occur in other observational or quasi-experimental studies examining the psychological impact of playing violent video games. As a result, the findings offer a more compelling and rigorous assessment of the potential effects of video game violence on laboratory measures of empathy and emotional reactivity.

2. The analyses in this study were conducted and reported transparently, following a registered analysis plan, and overall appear to be satisfactory. In particular, the authors' use of Bayesian hypothesis tests helped quantify the strength of the evidence for the lack of an effect.

Weaknesses:

1. One major concern is the contrast between the limited ecological validity of the design and the claims that this study can contribute to the controversy of the larger societal and psychological effects of video game use. There are several aspects of the design that limits its ecological validity, and thus its capacity to speak to this controversy:

– The sole inclusion of males between 18-35, who do not regularly play video games and have not played one of the most popular video games of all time restricts generalization to a very specific population.

– The short amount of exposure, 7 hours over two weeks, is low considering that according to various surveys, gamers play on average ~ 8 hours per week.

– The game played by the participants lacked many of the core features of popular video games such as immersion in a complex context/story, emotional investment and role-playing, progression and conflict between objectives, an internal motivation to play, etc.

– The link between laboratory measures of empathy and emotional reactivity and real-world behaviour is not supported by strong evidence and as such it is unclear what changes (or no changes) on these measures would imply for the impact of violent video games on psychological functioning.

Some of these limitations are underlined by the authors. However, the claim that these results can provide evidence on whether video games in everyday life do or do not "desensitize others to the plight of others" is a generalization beyond what the study design allows.

2. While the authors claim to support the absence of the effect of video games using Bayesian hypotheses tests, it is important to note that such tests are not immune to incorrect conclusions and misleading evidence, as they depend on the selection of prior distributions, model assumptions, and the choice of the strength of evidence threshold. This is particularly relevant for small effects and sample sizes (see Schönbrodt and Wagenmakers, 2017, Psychonomic Bulletin and Review).

From a frequentist point of view, with a two-sample one-tailed t-test (α=0.05), this study design has 80% power to detect moderate effects (d ~ 0.5) and will fail to detect small (d~0.2) effects most of the time. To justify the selected sample size and nuance their interpretations, the authors should consider the practical implications of different effect sizes and discuss the possibility that smaller effect sizes may exist but remain undetected and whether they believe these smaller effect sizes would have any practical importance.

Relatedly, the authors should consider the reliability of their measures and whether the effect size of interest was larger than the minimally detectable change allowed by these measures. This is particularly important given previous suggestions that task-evoked univariate fMRI responses have relatively poor reliability (Elliott et al., 2020, Psychological Science).

– The fMRI analyses focus on specific hypotheses tested in regions of interest. The authors might consider a wider exploration of the data that could help paint a clearer picture of the results. Differences in brain regions other than the specific ROIs could be interesting and provide hypotheses for further research. I'm not suggesting the authors reframe the whole manuscript based on such exploratory analyses but an additional section/appendix presenting the whole-brain results of the contrast of interest (Group x Intensity) in both tasks could be useful. It could also be interesting to consider changes in multivariate patterns of activity, for example, by assessing the pre-post response of previously developed multivariate whole-brain markers of empathic responses (Ashar et al., Neuron, 2017, markers shared with the paper).

– Please report all the parameters of the power analysis (α, power, specific effect size used and its justification, parametric distributions used, etc.) and consider adding a Bayes factor design analysis by performing Monte Carlo simulations that could inform on the capacity of Bayes factors to provide correct evidence with this sample size for the effect size of interest (see Schönbrodt, F. D., and Wagenmakers, E. J. (2018). Bayes factor design analysis: Planning for compelling evidence. Psychonomic bulletin and review, 25(1), 128-142).

– Please report the descriptive statistics of the amount of reward (or the number of photos/murders) in each group. If there is a large variation in gaming behaviour, changes in the dependent variables according to the number of murders could be considered in an exploratory analysis.

– Reporting standardized effect sizes with the Bayes factors would help the reader assess the results.

– I suggest changing the title of the paper to remove the play on words and to better reflect the nature of the study if possible (short intervention, fMRI measures, etc.).

– Please specify whether violence from other characters was present for people playing the photograph version of the game.

– I have not found the "Data Availability Statement" required as per *eLife* policy and the data shared on the OSF (behavioural and ROIs extracted data) alone cannot be used to perform several analyses reported in the manuscript. The authors should consider maximally sharing the data analyzed in the manuscript and adequately documenting the data to favour reuse.

– The manuscript indicates that game images will be shared on request, but these seem to be available on the OSF repository of the study.

– It would be helpful to state the sample size in the abstract and at the beginning of the Results section.

*Reviewer #3 (Recommendations for the authors):*

The study has several strengths and raises important questions on the broader question of the psychological impact of violent video games. The use of a longitudinal design while controlling the amount of exposure to violent video games represents an advantage over past studies. It would be useful to know how the decision on the frequency and dosage of video games (2 weeks, 1 hour each day) was arrived at. It is not clear if this decision was governed by feasibility constraints alone, or if is there any prior literature that points to this dosage. Second, the impact of playing violent (/non-violent) video games is measured using the empathy for pain paradigm in this study. It would be useful to note the test-retest reliability of the neural response in the empathy for pain paradigm. Relatedly, is there any data to suggest that this neural response is sensitive to change in any external factors?

1) It would be good to justify the choice of the smoothing kernel size.

2) It is not entirely clear why the signal changes were extracted from ROIs only after thresholding the statistical maps. Arguably, if the hypotheses are about specific regions, and not about peak voxels that are significant at the level of the whole brain, then extracting the relevant metric from the entire ROI is likely to be more informative.

3) It is not clear whether the ROIs for the empathy-for-pain metric were based on the thresholded group maps from the first session only – or were they different between sessions?

4) The use of two paradigms to measure the response to video games allows one to test the generalisability of the results. It would be useful to note if there is any relationship between the key metrics from these two paradigms, i.e. do individuals who show a greater empathy for pain (neurally/behaviourally) also show greater emotional reactivity?

---

## [Author Response]

Essential revisions:1. Abstract. Given the potentially damaging nature of conclusions that there is no impact of VGGs on emotional response in general, we would like it to be mentioned in the abstract the groups/settings to which this may not generalise.

We fully understand this concern, and have now added the following text to the abstract:

“While VVGs might not have a discernible effect on the investigated subpopulation within our carefully controlled experimental setting, our results cannot preclude that effects could be found in settings with higher ecological validity, in vulnerable subpopulations, or after more extensive VVG play.”

2. Generalisation. The authors already discuss that the findings may not generalise to clinical groups, younger groups, or longer/different types of gameplay. However, all reviewers felt that greater emphasis should be placed on this issue, especially due to the danger associated with misinterpretation of the present dataset. They would therefore like to see greater discussion of:(a) The authors picked 18-35 year old males who had not played VVGs at least 12 months before testing, nor played Grand Theft Auto ever before. This is one of the most popular games of all time and it presumably highly constrained their sample. We understand why they did this with respect to control, but wonder whether they also therefore preselect for a certain phenotype who is less likely to desensitise. We feel that the authors should discuss that the findings may not generalise to those who would actually choose to play these games (see below for details).

We thank you for highlighting this important limitation. Indeed, it is a highly relevant question if our selection criteria constrained our sample to a subpopulation that is less susceptible to the effects of VVGs, for example due particularly high trait empathy. Therefore, we have now conducted an additional exploratory analysis, where we compared the empathic traits (measured with the QCAE) of our sample to those of an independent sample of 18-35 year old males who were not preselected according to our videogame-related criteria. These analyses and their results are described in the new sections Methods: Post-hoc analyses: Sample comparability (lines 603-614) and Results: Post-hoc analyses: Sample comparability (lines 215-228). We found substantial evidence that the subpopulation from which we drew our sample exhibited similar levels of empathy on all five dimensions of the QCAE. However, we cannot preclude that our sample was particularly resistant to VVG effects due to other, untested characteristics. Further research is needed to assess if our results generalize to samples with other characteristics that may be more representative for the general population.

We have now also added the following text to the discussion to emphasize this issue (lines 312-319):

“To increase experimental control, we restricted our sample to young adult males who had minimal prior exposure to violent video games. It is possible that, due to this strict preselection criterion, our sample was drawn from a subpopulation that is particularly resistant to desensitization. An exploratory analysis provided strong evidence that our selection criterion did not result in particularly high levels of trait empathy in our sample, though. However, we cannot preclude that our sample was particularly resistant to VVG effects due to other, untested characteristics. Further research is needed to assess if our results generalize to samples with other characteristics that may be more representative for the general population.”

(b) They discuss that effects may differ with more video game exposure, but it may be worth properly considering that the level of exposure in the present study is arguably lower than among a large number of gamers – for whom the impact of VVGs is of greatest concern. It would also improve the manuscript to include discussion of how the quantity of exposure was decided. It is not clear if this decision was governed by feasibility constraints alone, or is there any prior literature that points to this dosage.

We thank you for these valuable recommendations. We have now adapted the discussion to more strongly emphasize the limitation that our experimental VVG is much lower than levels among habitual gamers (lines 299-306):

“Our experimental design ensured that participants of the experimental group were exposed to a substantial amount of violent gameplay during gaming sessions (participants “killed” on average 2000 other characters in a graphically violent way). However, the overall exposure to virtual violence was still very low when compared to the amount that is possible with extended VVG play in a typical everyday life setting. During our experiment, participants played for 7 hours over the course of two weeks. However, habitual gamers play, on average, as many as 16 hours in the same time frame (Clement, 2021; Statista Research Department, 2022). Our results cannot preclude that longer and more intense exposure to VVGs could have negative causal effects on empathy.”

Moreover, we have added an explanation regarding the quantity of exposure in the Methods section (lines 517-523):

“Due to the lack of other studies implementing a randomized experimental prospective design (except for Kühn et al. (2019), published while data collection was already on-going), there were no benchmarks for the amount and frequency of video game exposure for our study. We chose our regimen (seven one-hourly sessions over two weeks) as we considered this a substantial yet still feasible amount of exposure. We note though that the number of sessions, playing time per session, and total playing time were considerably higher than in previous studies reporting VVG effects on empathy (Arriaga et al., 2011; Carnagey et al., 2007; Engelhardt et al., 2011; Hasan et al., 2013).”

(c) The game played by the participants lacked many of the core features of popular video games such as immersion in a complex context/story, emotional investment and role-playing, progression and conflict between objectives, an internal motivation to play, etc.

Thank you for this comment. In response, we would like to clarify our theoretical rationale for this design choice: our main aim was to assess the effect of the mere exposure to extreme virtual violence, compared to a closely matched control condition with no such violence, on our dependent variables. Our two experimental conditions were highly comparable on all other dimensions other than violent content, such as (lack of) story, immersion, progression, etc. – allowing us to clearly identify the behavioral effects that are due to the violent content. Thus, our results provide evidence that violent content alone (in the amount presented in our experiment) is not enough to desensitize players to real-world pain and violence.

While we maintain that this design choice helped us to address the main aim with highest rigor, we agree that it is possible that the *combination* of violent content and strong immersion would have a different effect on empathy. We address this possibility in the discussion (lines 320-326):

“To maximize the amount of violence that participants would be exposed to (and commit) in the game, we restricted the game’s objective to killing other characters, and incentivized this behavior with monetary rewards. This might have reduced the ecological validity of our operationalization of gaming, and it is possible that bigger effects could be seen when violent gameplay is more intrinsically motivated or embedded in a more immersive overall game play; i.e. individuals who want to play the game out of intrinsic interest may be differently affected than those that have merely accepted to play it as part of an experiment. However, this does not invalidate the present evidence that the mere exposure to virtual violence for seven hours over two weeks is not sufficient to decrease empathy.”

(d) The link between laboratory measures of empathy and emotional reactivity and real-world behaviour is not supported by strong evidence and as such it is unclear what changes (or no changes) on these measures would imply for the impact of violent video games on psychological functioning.

This is a valid concern, but we would argue that this applies to any kind of experimental laboratory-based research, whether neuroscientific or behavioral-psychological. By definition, laboratory research will have lower external validity than field-based investigations. The latter however will obviously suffer from lower internal validity. Moreover, since our main interest are the neural bases of VVG-related changes in empathy, methodological constraints to perform neuroscience measures “in the field” would prevent us from investigating most if not all questions we focused on. We do not think however that the reviewers intend to suggest that tightly controlled laboratory research has no place and validity in advancing our understanding of real-life phenomena, for VVG but also more generally. We thus interpreted this comment as an invitation to emphasize the well-known, but admittedly often ignored strengths and limitations of laboratory vs. field-based research. We did this as follows, in the discussion (lines 327-337):

“It should be noted that there are few studies that connect laboratory-based experimental investigations of empathy and emotional reactivity to real-world behavior and its measures. There are indications, however, that neuroscientifc empathy measures similar to the ones used here predict individual social behavior (e.g., donation, helping or care-based behavior, e.g. Ashar et al., 2017; Hein et al., 2010; Hartmann et al., 2022; Tomova et al., 2017), and that they are also validated by their predictivity of mental or preclinical disorders characterized by deficits in empathy (e.g. Bird et al., 2010; Lamm et al., 2016, for review). That said, it is obvious that future research is needed that bridges and integrates laboratory and field-based measures and approaches, in order to inform us how changes (or their absence) in neural responses induced by VVG play are connected to real-life social emotions and behaviors (see Stijovic et al., 2023, for a recent example illustrating, in the domain of social isolation research, how a combined lab- and field based study can be directly informed by prior laboratory-based neuroscience findings).”

3. Mechanism. The authors consider how it is important to dissociate persistent effects of VGGs from short term effects such as arousal or priming. This could do with some greater clarification for the reader. Specifically, while we must dissociate effects half an hour later from those that persist for longer, one could easily describe longer-term impacts that operate due to arousal or priming mechanisms. In clarifying the argument, it would be helpful to know whether these previous experimental studies have found influences, or no influences, of the gaming. Further details are provided below.

We thank you for highlighting that this point needed clarification. We have now clarified this aspect in the introduction (lines 58-71):

“The existing experimental studies have nearly always used VVGs as an experimental manipulation shortly before measuring the outcomes of interest (Arriaga et al., 2011; Bushman and Anderson, 2009; Carnagey et al., 2007; Engelhardt et al., 2011; Guo et al., 2013; Staude-Müller et al., 2008). While these studies consistently report evidence for a desensitizing effect of violent games, they cannot disentangle the immediate effects of VVG play from those that have a persistent, long-term impact on individuals. Immediate VVG effects may encompass a wide range of processes, such as priming (Bushman, 1998), as well as stress-like responses such as increases in active fear and aggressive behaviors (Fanselow, 1994; Mobbs et al., 2007, 2009), that include generally increased sympathetic activity, release of stress hormones, heightened activation of involved brain structures, and cognitive-affective responses (e.g., deep reflection on the seen content, and changes in emotions and mood). Such responses can persist on a time-scale of minutes to hours after aversive events such as VVG exposure, and have been shown to negatively affect social behavior (Nitschke et al., 2022). It is important to distinguish these immediate effects from longer-term adaptations that occur over days or weeks, such as habituation or memory consolidation processes. The General Aggression Model predicts that the repeated exposure to violence in the positive emotional context of video games leads to the gradual extinction of aversive reactions, resulting in the long-term desensitization of players to real-world violence (Bushman and Anderson, 2009).”

4. Analysis specifics. More details about the parameters in the Bayesian tests and power calculations are requested. Additionally, despite the authors' truly commendable sample size, it must be noted that this is still insufficiently powered to detect small effects. And might a small effect with such implications still be important?

Thank you for highlighting that this was missing from the paper. We have now added a section on the power analysis at the beginning of the Methods section (lines 388-407):

“We planned to collect data from 90 participants. We derived this sample size from a power analysis based on VVG effect sizes reported in the meta-analysis of Anderson et al. (2010). The authors estimated the size of the negative VVG effect on empathy/desensitization to be r = 0.194, 95% CI = [0.170, 0.217], which corresponds to Cohen’s d = 0.396, 95% CI = [0.345, 0.445], representing a small-to-medium effect. We chose d = 0.300 as the minimum effect size for which we wanted to achieve a power of 0.80, to ensure that we would have enough power even if the reported effect size was overestimated. Note that thus, the effect size we used was even smaller than the lower bound reported in Anderson et al. (2010). We performed the power analysis using the software G*Power 3.1.9.2. (Faul et al., 2007), calculating the required sample size to achieve a power of 0.8 for the interaction in a 2-by-2 within-between design ANOVA, assuming a medium correlation of 0.5 between repeated measures, and using the conventional α error level of 0.05. This resulted in a required sample size of 90. Using such a sample size, the achieved power for the effect size reported in Anderson et al. (2010), as well as its lower and upper bound, was as follows: for d = 0.345, achieved power = 0.901; for d = 0.396, achieved power = 0.960; for d = 0.445, achieved power = 0.986.

Please note that while this power analysis was based on a frequentist analysis framework, we are reporting Bayesian analyses here. However, we considered this power analysis to be a sensible benchmark for the sample size needed to answer our research questions. See Results: Post-hoc analyses: Bayesian design analysis for a Post-hoc Bayesian design analysis that provides more information on the size of effects that could be detected with our sample size using Bayesian analyses.”

Following the second reviewer’s recommendation, we have now also conducted a post-hoc Bayes Factor design analysis by means of a Monte Carlo simulation experiment. This analysis is reported in the new sections Results: Post-hoc analyses: Bayesian design analysis (lines 242-265) and Results: Post-hoc analyses: Bayesian design analysis *(*lines 635-653*)*.

In summary, the simulation experiment suggested that our behavioral analyses were well enough powered to differentiate between the absence and presence of a medium-to-small effect of d = 0.3. For smaller effects, such as d = 0.2, the a priori power of our behavioral analyses was not optimal, as it would have been likely that we would have obtained an inconclusive result (⅓ < BF < 3) even in the presence of a true effect of that size. However, given that we obtained evidence for the null hypothesis (BF < 1/3) in all relevant Bayes Factor tests on our behavioral data, our results speak strongly against the presence of such an effect.

Regarding our neural analyses, given the low correlation between repeated measurements/test-retest reliability, the Bayesian power of our fMRI analyses should be regarded as low (also see our response regarding test-retest reliability below). Taken alone, we would not consider them convincing evidence against the presence of a VVG effect. However, together with our behavioral results, they suggest that VVG effects, if they exist, can be expected to be smaller than the ones reported in the literature.

To address these points, we have added the following section to the discussion (lines 338-345):

“Our study was designed to reliably detect an effect size of d = 0.3, an effect even smaller than the lower estimate for VVG effects on empathy reported in Anderson et al. (2010). Our results provide substantial evidence that effects of this magnitude are not present in settings similar to our experimental design. These arguments notwithstanding, it needs to be noted that future studies with higher power may detect still smaller effects. Considering the high prevalence of VVG, even such small effects could be of high societal relevance (Funder and Ozer, 2019). For now, based on the current design and data, we can conclude that experimental long-term VVG effects on empathy are unlikely to be as large as previously reported.”

5. Test. The impact of playing violent (/non-violent) video games is measured using the empathy for pain paradigm in this study. It would be useful to note the test-retest reliability of the neural response in the empathy for pain paradigm.

We have now added estimates of the test-retest reliability of our measurements to the manuscript, in the new sections Results: Post-hoc analyses: Test-retest reliability (lines 229-241) and Methods: Post-hoc analyses: Test-retest reliability (lines 615-634).

For our behavioral measurements, reliability was high to very high (⍴ ~ 0.7 for painfulness ratings, ⍴ ~ 0.9 for unpleasantness ratings). For neural responses, reliability was close to zero, though (⍴ ~ 0 for the three considered ROIs). To the best of our knowledge, our study was the first one to present the empathy for pain paradigm to the same sample of participants after a longer time frame. Thus, this surprising result provides valuable information on the limitations of this task respectively the neural measurements acquired in it, and certainly demands further research to investigate the factors influencing fMRI reliability.

We would like to emphasize, though, that a high test-retest reliability is not a precondition for the valid testing of group-level effects. For a group-level effect to be testable, it is only necessary that the mean of the dependent variable is consistently affected by the independent variable. It is not necessary that participants who show an above average level in the DV in one session also show an above average level in the second session, and vice versa. Otherwise, there would also be no point in independent-sample designs. Indeed, it has recently been discussed that highly robust cognitive tasks are bound to exhibit low test-retest reliability, as robust tasks are often characterized by low interindividual variation, and thus leave only little variance that can be explained by participant traits (Hedge et al., 2018). However, it must also be noted that low reliability does lead to lower power of repeated measures designs. As discussed in our previous response on power, the low reliability of the measured neural responses has resulted in suboptimal power of our tests on fMRI data.

We have added discussion of these points to the manuscript (lines 353-371):

“Lastly, and somewhat surprisingly, we found that the test-retest reliability of our neural covariates of empathy for pain were close to zero for all investigated ROIs. Knowing that an individual’s neural empathic response (BOLD activity for seeing somebody else in pain vs. in no pain) was above or below average in the first session provides little to no information about their relative response in the second session. To the best of our knowledge, our study was the first one to present the empathy for pain paradigm to the same sample of participants after a longer time frame. Thus, this surprising result provides valuable information on the limitations of this task respectively the neural measurements acquired in it, and certainly demands further research to investigate the factors influencing fMRI reliability (see also Elliott et al., 2020; Kragel et al., 2021). We would like to emphasize, though, that a high test-retest reliability is not a precondition for the valid testing of group-level effects. For a group-level effect to be testable, it is only necessary that the mean of the dependent variable is consistently affected by the independent variable. It is not necessary that participants who show an above average level in the DV in one session also show an above average level in the second session, and vice versa. Otherwise, there would also be no point in independent-sample designs. Indeed, it has recently been discussed that highly robust cognitive tasks are bound to exhibit low test-retest reliability, as robust tasks are often characterized by low interindividual variation, and thus leave only little variance that can be explained by participant traits (Hedge et al., 2018). However, it must also be noted that low reliability does lead to lower power of repeated measures designs, and that the low reliability of the measured neural responses has resulted in suboptimal power of our tests on fMRI data.”

Relatedly, is there any data to suggest that this neural response is sensitive to change in any external factors?

The empathy-for-pain paradigm that we used is well established, and the empathic neural response has been shown to be sensitive to be affected by various types of external factors. Among the factors that have been reported to modulate the response are ingroup-outgroup bias (Hein et al., 2010), perspective-taking (Lamm et al., 2007), placebo analgesia (Rütgen et al., 2015), and antidepressants (Rütgen et al., 2019). Especially the work using placebo analgesia, an intervention that traditionally has rather low effect sizes on first-hand pain, suggests that the lack of effects by the VVG is not due to the insufficient malleability of empathy and its neural bases, as operationalized and measured here. We clarified this point as follows in the discussion (lines 346-352):

It may be argued that the empathy for pain paradigm and the associated behavioral and neural responses are so robust and resistant to changes by external factors that this may explain the lack of evidence for the effects of VVG play. This argument however would contradict a wealth of findings illustrating malleability of empathic responses using this and related designs, including with placebo analgesia (Rütgen et al., 2021; Rütgen, Seidel, Riečanský, et al., 2015; Rütgen, Seidel, Silani, et al., 2015), an intervention that usually shows low to moderate effect sizes as well (see e.g. Hein and Singer, 2008; Jauniaux et al., 2019; Claus Lamm et al., 2019; for review).Furthermore, the use of two paradigms to measure the response to video games allows one to test the generalisability of the results. It would be useful to note if there is any relationship between the key metrics from these two paradigms, i.e. do individuals who show a greater empathy for pain (neurally/behaviourally) also show greater emotional reactivity?

Thank you for this interesting suggestion. We have now added analyses of cross-task correlations to the manuscript in the new sections *Results: Post-hoc analyses: cross-task correlations* (lines 266-272) and *Results: Post-hoc analyses: cross-task correlations (*lines 654-662*)*. They are also reproduced at the end of this response letter. In short, we found small-to-medium correlations (.227 –.280) between the behavioral measures of the empathy for pain task and the emotional reactivity task. However, we could not observe any noteworthy correlations between the neural measurements. This might reflect that the two tasks tap into two distinct, yet related processes: while one measures empathy for pain, the other measures responses to visually upsetting situations.

Reviewer #1 (Recommendations for the authors):The authors provide compelling support for the idea that playing VGGs yields no influence on one's ratings of painfulness or unpleasantness when observing others in pain, or unpleasantness ratings when observing videos of violence – in their particular sample of interest. fMRI data showed no difference in regions responsive to others' pain, or those responsive to observed violence, as a function of VGG.I was very impressed with this manuscript. It was a joy to read due to its clarity of expression and simple paradigm answering a clear theoretical question. I found the data compelling, in an impressively large sample, and believe the findings will inform this thorny debate and be of interest to a wide readership. I was especially impressed with the crystal clear manner in which the authors highlighted previous work, along with the questions that remain unanswered and how this study will answer them. Such a rewarding read, and notably unusual for authors to make the life of the reader so easy in this manner. Thank you.

We thank the reviewer very much for these kind words and positive feedback. We are very happy that we were able to communicate our findings in a clear and easily readable way.

I just have a couple of suggestions for improvement, relating to the underlying mechanism and generalisability.Mechanism. The authors consider how it is important to dissociate persistent effects of VGGs from short term effects such as arousal or priming. I think I have some idea what the authors mean but feel it could do with some greater clarification for the reader. Specifically, of course, we must dissociate effects half an hour later from those that persist for longer, and I understand the main point. However, one could easily describe longer-term impacts that operate due to arousal or priming mechanisms. I think it is important for the reader to outline the specific mechanistic account that would have short-term, but not long-term, impact. Presumably, we are talking some type of habituation/adaptation mechanism, whereby activation of structures such as the amygdala during gameplay means that they cannot be activated subsequently so readily? I don't quite follow this, especially given that presumably there is evidence of activation of these structures to observation of pain but it's the mediation by the gaming that differs. In clarifying the argument, it would be helpful to know whether these previous experimental studies have found influences, or no influences, of the gaming.

We thank you for highlighting that this point needed clarification. We have now clarified this aspect in the introduction (lines 58-71):

“The existing experimental studies have nearly always used VVGs as an experimental manipulation shortly before measuring the outcomes of interest (Arriaga et al., 2011; Bushman and Anderson, 2009; Carnagey et al., 2007; Engelhardt et al., 2011; Guo et al., 2013; Staude-Müller et al., 2008). While these studies consistently report evidence for a desensitizing effect of violent games, they cannot disentangle the immediate effects of VVG play from those that have a persistent, long-term impact on individuals. Immediate VVG effects may encompass a wide range of processes, such as priming (Bushman, 1998), as well as stress-like responses such as increases in sympathetic activity, release of stress hormones, heightened activation of involved brain structures, and cognitive-affective responses (e.g., deep reflection on the seen content, and changes in emotions and mood). Such responses can persist on a time-scale of minutes to hours after aversive events such as VVG exposure, and have been shown to negatively affect social behavior (Nitschke et al., 2022). It is important to distinguish these immediate effects from longer-term adaptations that occur over days or weeks, such as habituation or memory consolidation processes. The General Aggression Model predicts that the repeated exposure to violence in the positive emotional context of video games leads to the gradual extinction of aversive reactions, resulting in the long-term desensitization of players to real-world violence (Bushman and Anderson, 2009).”

Generalisation. The authors already discuss that the findings may not generalise to clinical groups, younger groups, or longer/different types of gameplay. Another dimension jumped out to me and I think the manuscript would benefit from its discussion. Specifically, they pick 18-35-year-old males who had not played VVGs at least 12 months before testing, and not ever played Grand Theft Auto. I understand why they did this with respect to control, but I wonder whether they also, therefore, preselect for a certain phenotype who is less likely to desensitise. If they have not chosen to play these types of game this perhaps means they do not receive the positive emotional reward of violence that some other participants may do. I imagine that for large swathes of the population the negative emotional response to observing death is so extreme that they would obtain no reward whatsoever from these games. These are possibly the individuals tested here, and their response could prove less relevant to understanding those who actually play them – and for whom this debate has practical implications. Those who receive the intrinsic positive reward, due to feelings of power or the like, may desensitise in a way that this sample do not. Therefore, while I think these findings are highly compelling in this sample, I think the authors should discuss that the findings may not generalise to those who would actually choose to play these games.

We thank you for highlighting this important limitation. Indeed, it is a highly relevant question if our selection criteria constrained our sample to a subpopulation that is less susceptible to the effects of VVGs, for example due particularly high trait empathy, and therefore high negative emotional response to depictions of death and violence. Therefore, we have now conducted an additional exploratory analysis, where we compared the empathic traits (measured with the QCAE) of our sample to those of an independent sample of 18-35 year old males who were not preselected according to our videogame-related criteria. These analyses and their results are described in the new sections Methods: Post-hoc analyses: Sample comparability (lines 603-614) and Results: Post-hoc analyses: Sample comparability (lines 215-228). We found substantial evidence that the subpopulation from which we drew our sample exhibited similar levels of empathy on all five dimensions of the QCAE. However, we cannot preclude that our sample was particularly resistant to VVG effects due to other, untested characteristics. Further research is needed to assess if our results generalize to samples with other characteristics that may be more representative for the general population.

We have now also added the following text to the discussion to emphasize this issue (lines 312-319):

“To increase experimental control, we restricted our sample to young adult males who had minimal prior exposure to violent video games. It is possible that, due to this strict preselection criterion, our sample was drawn from a subpopulation that is particularly resistant to desensitization. An exploratory analysis provided strong evidence that our selection criterion did not result in particularly high levels of trait empathy in our sample, though. However, we cannot preclude that our sample was particularly resistant to VVG effects due to other, untested characteristics. Further research is needed to assess if our results generalize to samples with other characteristics that may be more representative for the general population.”

Abstract. Given the potentially damaging nature of conclusions that there is no impact of VGGs on emotional response in general, I think it is wise to mention in the abstract the groups/settings to which this may not generalise.

We fully understand this concern, and have now added the following text to the abstract:

“While VVGs might not have a discernible effect on the investigated subpopulation within our carefully controlled experimental setting, our results cannot preclude that effects could be found in settings with higher ecological validity, on other tasks, in vulnerable subpopulations, or after more extensive VVG play.”

I note in Figure 2 that all trends are in predicted directions, although I guess the Bayesian stats suggest this is irrelevant. Worth a comment?

Indeed, while Figure 2 (especially Figure 2B) suggests a trend in the predicted direction, the Bayesian hypothesis tests reveal that this pattern is much better explained by random noise than by a systematic effect. According to the Bayes factor, the data are about 7.7 times more likely under a model without a VVG effect than under a model with a VVG effect (interaction Group*Session*Intensity). We have added a note regarding this to the caption of Figure 2.

Reviewer #2 (Recommendations for the authors):Lengersdorff and colleagues conducted a study to examine the impact of exposure to video-game violence on physiological (fMRI) and behavioral measures of empathic responses and emotional reactivity. In the experimental study, the authors initially assessed empathic responses and emotional reactivity in 89 participants at baseline. Then, roughly half of the participants were randomly assigned to either a violent video game condition or a control condition. During the intervention phase, participants in the violent condition played a modified video game that incentivized graphical acts of interpersonal violence for seven hours over two weeks. Meanwhile, participants in the control condition played a similar game but were asked to take pictures instead of committing violence. The findings of the study suggest that this intervention had no effect on brain responses or behavioral responses previously linked to empathy for pain in others, or on responses linked to emotional reactivity.The data from this study offer valuable insights into the impact of brief exposure to video game violence on laboratory measures of empathy and emotional reactivity and the results appear to be free of confounds. However, it should be noted that the study's ability to shed light on the broader impact of video games on psychological functioning outside of the laboratory is limited.Strengths:1. The authors were able to enhance the internal validity of their research by developing an experiment that included a well-crafted control condition, pre-post measures, and a carefully chosen participant population. This approach enabled them to minimize the influence of many confounding factors that frequently occur in other observational or quasi-experimental studies examining the psychological impact of playing violent video games. As a result, the findings offer a more compelling and rigorous assessment of the potential effects of video game violence on laboratory measures of empathy and emotional reactivity.2. The analyses in this study were conducted and reported transparently, following a registered analysis plan, and overall appear to be satisfactory. In particular, the authors' use of Bayesian hypothesis tests helped quantify the strength of the evidence for the lack of an effect.

We would like to thank the reviewer very much for their positive and constructive feedback, in particular regarding the limitations of our results regarding generalizability, and how they can be addressed. Please find our point-by-point response below.

Weaknesses:1. One major concern is the contrast between the limited ecological validity of the design and the claims that this study can contribute to the controversy of the larger societal and psychological effects of video game use. There are several aspects of the design that limits its ecological validity, and thus its capacity to speak to this controversy:– The sole inclusion of males between 18-35, who do not regularly play video games and have not played one of the most popular video games of all time restricts generalization to a very specific population.

We thank you for highlighting this important limitation. Indeed, it is a highly relevant question if our selection criteria constrained our sample to a subpopulation that is less susceptible to the effects of VVGs, for example due particularly high trait empathy, and therefore high negative emotional response to depictions of death and violence. Therefore, we have now conducted an additional exploratory analysis, where we compared the empathic traits (measured with the QCAE) of our sample to those of an independent sample of 18-35 year old males who were not preselected according to our videogame-related criteria. These analyses and their results are described in the new sections Methods: Post-hoc analyses: Sample comparability (lines 603-614) and Results: Post-hoc analyses: Sample comparability (lines 215-228). We found substantial evidence that the subpopulation from which we drew our sample exhibited similar levels of empathy on all five dimensions of the QCAE. However, we cannot preclude that our sample was particularly resistant to VVG effects due to other, untested characteristics. Further research is needed to assess if our results generalize to samples with other characteristics that may be more representative for the general population.

We have now also added the following text to the discussion to emphasize this issue (lines 312-319):

“To increase experimental control, we restricted our sample to young adult males who had minimal prior exposure to violent video games. It is possible that, due to this strict preselection criterion, our sample was drawn from a subpopulation that is particularly resistant to desensitization. An exploratory analysis provided strong evidence that our selection criterion did not result in particularly high levels of trait empathy in our sample, though. However, we cannot preclude that our sample was particularly resistant to VVG effects due to other, untested characteristics. Further research is needed to assess if our results generalize to samples with other characteristics that may be more representative for the general population.”

– The short amount of exposure, 7 hours over two weeks, is low considering that according to various surveys, gamers play on average ~ 8 hours per week.

We have now adapted the discussion to more strongly emphasize the limitation that our experimental VVG is much lower than levels among habitual gamers (lines 299-306):

“Our experimental design ensured that participants of the experimental group were exposed to a substantial amount of violent gameplay during gaming sessions (participants “killed“ on average 2000 other characters in a graphically violent way). However, the overall exposure to virtual violence was still very low when compared to the amount that is possible with extended VVG play in a typical everyday life setting. During our experiment, participants played for 7 hours over the course of two weeks. However, habitual gamers play, on average, as many as 16 hours in the same time frame (Clement, 2021; Statista Research Department, 2022). We thus would like to stress that our results remain agnostic as to how longer, more extended, and more intense exposure to VVGs will impact empathy.”

– The game played by the participants lacked many of the core features of popular video games such as immersion in a complex context/story, emotional investment and role-playing, progression and conflict between objectives, an internal motivation to play, etc.

Thank you for this comment. In response, we would like to clarify our theoretical rationale for this design choice: our main aim was to assess the effect of the mere exposure to extreme virtual violence, compared to a closely matched control condition with no such violence, on our dependent variables. Our two experimental conditions were highly comparable on all other dimensions other than violent content, such as (lack of) story, immersion, progression, etc. – allowing us to clearly identify the behavioral effects that are due to the violent content. Thus, our results provide evidence that violent content alone (in the amount presented in our experiment) is not enough to desensitize players to real-world pain and violence.

While we maintain that this design choice helped us to address the main aim with highest rigor, we agree that it is possible that the combination of violent content and strong immersion would have a different effect on empathy. We address this possibility in the discussion (lines 320-326):

“To maximize the amount of violence that participants would be exposed to (and commit) in the game, we restricted the game’s objective to killing other characters, and incentivized this behavior with monetary rewards. This might have reduced the ecological validity of our operationalization of gaming, and it is possible that bigger effects could be seen when violent gameplay is more intrinsically motivated or embedded in a more immersive overall game play; i.e. individuals who want to play the game out of intrinsic interest may be differently affected than those that have merely accepted to play it as part of an experiment. However, this does not invalidate the present evidence that the mere exposure to virtual violence for seven hours over two weeks is not sufficient to decrease empathy.”

– The link between laboratory measures of empathy and emotional reactivity and real-world behaviour is not supported by strong evidence and as such it is unclear what changes (or no changes) on these measures would imply for the impact of violent video games on psychological functioning.

This is a valid concern, but we would argue that this applies to any kind of experimental laboratory-based research, whether neuroscientific or behavioral-psychological. By definition, laboratory research will have lower external validity than field-based investigations. The latter however will obviously suffer from lower internal validity. Moreover, since our main interest are the neural bases of VVG-related changes in empathy, methodological constraints to perform neuroscience measures “in the field” would prevent us from investigating most if not all questions we focused on. We do not think however that the reviewers intend to suggest that tightly controlled laboratory research has no place and validity in advancing our understanding of real-life phenomena, for VVG but also more generally. We thus interpreted this comment as an invitation to emphasize the well-known, but admittedly often ignored strengths and limitations of laboratory vs. field-based research. We did this as follows, in the discussion (lines 327-337):

“It should be noted that there are few studies that connect laboratory-based experimental investigations of empathy and emotional reactivity to real-world behavior and its measures. There are indications, however, that neuroscientifc empathy measures similar to the ones used here predict individual social behavior (e.g., donation, helping or care-based behavior, e.g. Ashar et al., 2017; Hein et al., 2010; Hartmann et al., 2022; Tomova et al., 2017), and that they are also validated by their predictivity of mental or preclinical disorders characterized by deficits in empathy (e.g. Bird et al., 2010; Lamm et al., 2016, for review). That said, it is obvious that future research is needed that bridges and integrates laboratory and field-based measures and approaches, in order to inform us how changes (or their absence) in neural responses induced by VVG play are connected to real-life social emotions and behaviors (see Stijovic et al., 2023, for a recent example illustrating, in the domain of social isolation research, how a combined lab- and field based study can be directly informed by prior laboratory-based neuroscience findings).”

Some of these limitations are underlined by the authors. However, the claim that these results can provide evidence on whether video games in everyday life do or do not "desensitize others to the plight of others" is a generalization beyond what the study design allows.2. While the authors claim to support the absence of the effect of video games using Bayesian hypotheses tests, it is important to note that such tests are not immune to incorrect conclusions and misleading evidence, as they depend on the selection of prior distributions, model assumptions, and the choice of the strength of evidence threshold. This is particularly relevant for small effects and sample sizes (see Schönbrodt and Wagenmakers, 2017, Psychonomic Bulletin and Review).

We fully agree with the reviewer that Bayesian approaches are not a panacea against incorrect conclusions, and that the informed reader should always question if they find the choices of prior distributions and model assumptions sensible. This is why we describe the models and their priors in large detail in the methods section and relevant appendices. Regarding the choice of threshold, we see it as one of the great strengths of the Bayes factor that it is easily interpretable on a continuous scale, and that every reader can decide for themselves if they are convinced by the exact Bayes factor reported. To further emphasize this, we have added the following to the introduction (lines 102-104):

“We followed the convention to report a BF > 3 as evidence for the alternative hypothesis, a BF < 1/3 as evidence for the null hypothesis, and a BF in the interval [1/3, 3] as inconclusive evidence for either hypothesis (Kass and Raftery, 1995; Keysers et al., 2020). We would like to emphasize, though, that the Bayes factor provides an easily interpretable continuous quantification of the evidence for and against hypotheses, and that a strict categorization of BFs into evidence for and against hypotheses is not necessary.”

From a frequentist point of view, with a two-sample one-tailed t-test (α=0.05), this study design has 80% power to detect moderate effects (d ~ 0.5) and will fail to detect small (d~0.2) effects most of the time. To justify the selected sample size and nuance their interpretations, the authors should consider the practical implications of different effect sizes and discuss the possibility that smaller effect sizes may exist but remain undetected and whether they believe these smaller effect sizes would have any practical importance.

We thank the reviewer for their comments on power, and in general for their comments regarding the actual strength of our evidence. We now address these at several points throughout our manuscript:

First, we have added a detailed explanation of our power analysis at the beginning of the Methods section (lines 388-407):

“We planned to collect data from 90 participants. We derived this sample size from a power analysis based on VVG effect sizes reported in the meta-analysis of Anderson et al. (2010). The authors estimated the size of the negative VVG effect on empathy/desensitization to be r = 0.194, 95% CI = [0.170, 0.217], which corresponds to Cohen’s d = 0.396, 95% CI = [0.345, 0.445], representing a small-to-medium effect. We chose d = 0.300 as the minimum effect size for which we wanted to achieve a power of 0.80, to ensure that we would have enough power even if the reported effect size was overestimated. Note that thus, the effect size we used was even smaller than the lower bound reported in Anderson et al. (2010). We performed the power analysis using the software G*Power 3.1.9.2. (Faul et al., 2007), calculating the required sample size to achieve a power of 0.8 for the interaction in a 2-by-2 within-between design ANOVA, assuming a medium correlation of 0.5 between repeated measures, and using the conventional α error level of 0.05. This resulted in a required sample size of 90. Using such a sample size, the achieved power for the effect size reported in Anderson et al. (2010), as well as its lower and upper bound, was as follows: for d = 0.345, achieved power = 0.901; for d = 0.396, achieved power = 0.960; for d = 0.445, achieved power = 0.986.

Please note that while this power analysis was based on a frequentist analysis framework, we are reporting Bayesian analyses here. However, we considered this power analysis to be a sensible benchmark for the sample size needed to answer our research questions. See Results: Post-hoc analyses: Bayesian design analysis for a Post-hoc Bayesian design analysis that provides more information on the size of effects that could be detected with our sample size using Bayesian analyses.”

Here we would like to emphasize again that our study was designed to detect an effect that is even smaller (d = 0.300) than the lower bound of estimates reported in the meta-analysis of Anderson et al. (d = 0.345).

Second, following the reviewer’s suggestion, we have now also conducted a post-hoc Bayes Factor design analysis by means of a Monte Carlo simulation experiment. This analysis is reported in the new sections Results: Post-hoc analyses: Bayesian design analysis (lines 242-265) and Results: Post-hoc analyses: Bayesian design analysis (lines 635-653).

In summary, the simulation experiment suggested that our behavioral analyses were well enough powered to differentiate between the absence and presence of a medium-to-small effect of d = 0.3. For smaller effects, such as d = 0.2, the a priori power of our behavioral analyses was not optimal, as it would have been likely that we would have obtained an inconclusive result (⅓ < BF < 3) even in the presence of a true effect of that size. However, given that we obtained evidence for the null hypothesis (BF < 1/3) in all relevant Bayes Factor tests on our behavioral data, our results speak strongly against the presence of such an effect.

Regarding our neural analyses, given the low correlation between repeated measurements/test-retest reliability, the Bayesian power of our fMRI analyses should be regarded as low (also see our response regarding test-retest reliability below). Taken alone, we would not consider them convincing evidence against the presence of a VVG effect. However, together with our behavioral results, they suggest that VVG effects, if they exist, can be expected to be smaller than the ones reported in the literature.

Finally, we have added discussion of these points to the manuscript (lines 338-345):

“Our study was designed to reliably detect an effect size of d = 0.3, an effect even smaller than the lower estimate for VVG effects on empathy reported in Anderson et al. (2010). Our results provide substantial evidence that effects of this magnitude are not present in settings similar to our experimental design. These arguments notwithstanding, it needs to be noted that future studies with higher power may detect still smaller effects. Considering the high prevalence of VVG, even such small effects could be of high societal relevance (Funder and Ozer, 2019). For now, based on the current design and data, we can conclude that experimental long-term VVG effects on empathy are unlikely to be as large as previously reported.”

Relatedly, the authors should consider the reliability of their measures and whether the effect size of interest was larger than the minimally detectable change allowed by these measures. This is particularly important given previous suggestions that task-evoked univariate fMRI responses have relatively poor reliability (Elliott et al., 2020, Psychological Science).

We have now added estimates of the test-retest reliability of our measurements to the manuscript, in the new sections Results: Post-hoc analyses: Test-retest reliability (lines 229-241) and Methods: Post-hoc analyses: Test-retest reliability (lines 615-634). For our behavioral measurements, reliability was high to very high (⍴ ~ 0.7 for painfulness ratings, ⍴ ~ 0.9 for unpleasantness ratings). For neural responses, reliability was close to zero, though (⍴ ~ 0 for the three considered ROIs). To the best of our knowledge, our study was the first one to present the empathy for pain paradigm to the same sample of participants after a longer time frame. Thus, this surprising result provides valuable information on the limitations of this task respectively the neural measurements acquired in it, and certainly demands further research to investigate the factors influencing fMRI reliability.

We would like to emphasize, though, that a high test-retest reliability is not a precondition for the valid testing of group-level effects. For a group-level effect to be testable, it is only necessary that the mean of the dependent variable is consistently affected by the independent variable. It is not necessary that participants who show an above average level in the DV in one session also show an above average level in the second session, and vice versa. Otherwise, there would also be no point in independent-sample designs. Indeed, it has recently been discussed that highly robust cognitive tasks are bound to exhibit low test-retest reliability, as robust tasks are often characterized by low interindividual variation, and thus leave only little variance that can be explained by participant traits (Hedge et al., 2018). However, it must also be noted that low reliability does lead to lower power of repeated measures designs. As discussed in our previous response on power, the low reliability of the measured neural responses has resulted in suboptimal power of our tests on fMRI data.

We have added discussion of these points to the manuscript (lines 353-371):

“Lastly, and somewhat surprisingly, we found that the test-retest reliability of our neural covariates of empathy for pain were close to zero for all investigated ROIs. Knowing that an individual’s neural empathic response (BOLD activity for seeing somebody else in pain vs. in no pain) was above or below average in the first session provides little to no information about their relative response in the second session. To the best of our knowledge, our study was the first one to present the empathy for pain paradigm to the same sample of participants after a longer time frame. Thus, this surprising result provides valuable information on the limitations of this task respectively the neural measurements acquired in it, and certainly demands further research to investigate the factors influencing fMRI reliability (see also Elliott et al., 2020; Kragel et al., 2021). We would like to emphasize, though, that a high test-retest reliability is not a precondition for the valid testing of group-level effects. For a group-level effect to be testable, it is only necessary that the mean of the dependent variable is consistently affected by the independent variable. It is not necessary that participants who show an above average level in the DV in one session also show an above average level in the second session, and vice versa. Otherwise, there would also be no point in independent-sample designs. Indeed, it has recently been discussed that highly robust cognitive tasks are bound to exhibit low test-retest reliability, as robust tasks are often characterized by low interindividual variation, and thus leave only little variance that can be explained by participant traits (Hedge et al., 2018). However, it must also be noted that low reliability does lead to lower power of repeated measures designs, and that the low reliability of the measured neural responses has resulted in suboptimal power of our tests on fMRI data.”

– The fMRI analyses focus on specific hypotheses tested in regions of interest. The authors might consider a wider exploration of the data that could help paint a clearer picture of the results. Differences in brain regions other than the specific ROIs could be interesting and provide hypotheses for further research. I'm not suggesting the authors reframe the whole manuscript based on such exploratory analyses but an additional section/appendix presenting the whole-brain results of the contrast of interest (Group x Intensity) in both tasks could be useful. It could also be interesting to consider changes in multivariate patterns of activity, for example, by assessing the pre-post response of previously developed multivariate whole-brain markers of empathic responses (Ashar et al., Neuron, 2017, markers shared with the paper).

We agree with the reviewer that our data provide a wealth of possible additional analyses. However, we believe that the analyses we have presented so far are sufficient to exhaustively answer the research questions we wished to address in this specific paper.

The T-maps we have uploaded on Neurovault allow interested readers to investigate the whole-brain results of the contrasts of interest in both tasks. We have now emphasized this possibility in the manuscript (Appendix 5, lines 859-860):

“Other whole-brain results may be investigated using the T-map provided online (https://identifiers.org/neurovault.collection:13395).”

While we agree that it would be interesting to investigate changes in multivariate markers of empathy, we deem such complex additional analyses to be beyond the scope of the present paper. Given that we have now uploaded the complete fMRI datasets of each participant (see data availability statement), other researchers have the opportunity to investigate these interesting additional questions.

– Please report all the parameters of the power analysis (α, power, specific effect size used and its justification, parametric distributions used, etc.) and consider adding a Bayes factor design analysis by performing Monte Carlo simulations that could inform on the capacity of Bayes factors to provide correct evidence with this sample size for the effect size of interest (see Schönbrodt, F. D., and Wagenmakers, E. J. (2018). Bayes factor design analysis: Planning for compelling evidence. Psychonomic bulletin and review, 25(1), 128-142).

We thank the reviewer for these suggestions and kindly refer to the previous point of this letter, where we describe the steps we took to implement them.

– Please report the descriptive statistics of the amount of reward (or the number of photos/murders) in each group. If there is a large variation in gaming behaviour, changes in the dependent variables according to the number of murders could be considered in an exploratory analysis.

We now report these descriptive statistics at the beginning of the Results section (lines 111-115):

“On average, participants of the experimental group killed 2844.7 characters (SD = 993.9, Median = 2820, Minimum = 441, Maximum = 6815). Participants of the control group took an average of 3055.3 pictures of other characters (SD = 1307.5, Median = 3026, Minimum = 441, Maximum = 6815). Thus, as was the aim of our experimental design, each participant of the experimental group was exposed to a substantial number of violent acts in the video game.”

An exploratory analysis using the number of “kills” as independent variable revealed no additional effects in the behavioral data. We prefer not to report this additional analysis in the manuscript, to keep the focus on the confirmatory analyses.

– Reporting standardized effect sizes with the Bayes factors would help the reader assess the results.

Unfortunately, there are no established standardized effect sizes for the hierarchical Bayesian models reported here. However, for the behavioral results, the reported parameters are already on a meaningful scale, namely the 100-step VAS scale. We have now added additional help in interpretation to the text:

Lines 124-149:

“The posterior means of fixed effect parameters are listed in Table 1.A for Painfulness ratings, and Table 1.B for Unpleasantness ratings. As a manipulation check, we first tested whether painful stimuli led to increased painfulness and unpleasantness ratings, compared to non-painful stimuli. For both kinds of ratings, this test revealed very strong evidence (BF > 100) for an effect of Intensity, indicating that our paradigm was able to induce empathic responses in participants (see Figure 2A and B). The posterior mean of the regression parameter β of the factor Intensity was 27.86 for Painfulness ratings, and 17.48 for Unpleasantness ratings. Given our used factor coding, this means that the average difference in ratings between painful and non-painful stimuli was 2*27.86 = 55.72 points of the 100-point VAS for Painfulness ratings, and 2*17.48 = 55.72 points for Unpleasantness ratings.

We found evidence for the absence of a VVG effect on the painfulness ratings. Comparing a model where the fixed effect of Group*Time*Intensity could be negative to a model where the effect was set to zero resulted in a BF of 0.324. This means that the observed ratings were about 3.1 times more likely under the null hypothesis of no VVG effect than under the alternative hypothesis. When estimated without restrictions, the posterior mean of β for the interaction Group*Session*Intensity was -0.78. Given our factor codings, this means that the quantity [rating_Pain_ – rating_No Pain_]_Session 2 –_ [rating_Pain_ – rating_No Pain_]_Session 1_ (thus, the baseline corrected empathic response) was on average 1.56 points smaller in the experimental group than in the control group, on the 100-point VAS scale. However, note that the Bayesian hypothesis test suggests that a model with this interaction restricted to zero provides a better explanation of the data.

For the unpleasantness ratings, evidence for absence of a VVG effect was substantial. With a BF of 0.130, the observed data were about 7.7 times more likely under the null hypothesis of no VVG effect than under the alternative hypothesis. The posterior mean of β for the interaction Group*Session*Intensity was -0.45. Given our factor codings, this means that the quantity [rating_Pain_ – rating_No Pain_]_Session 2 –_ [rating_Pain_ – rating_No Pain_]_Session 1_ was on average 0.9 points smaller in the experimental group than in the control group. However, again note that the Bayesian hypothesis test suggests that a model without this interaction provides a better explanation of the data.”

Lines 157-176:

“The posterior means of fixed effect parameters of this model are listed in Table 2. As a manipulation check, we first tested whether participants experienced more unpleasantness in the emotional reactivity task while observing violent pictures compared to neutral pictures. We found very strong evidence (BF > 100) for this hypothesis, indicating that our paradigm was successful in inducing unpleasantness by violent imagery. The posterior mean of the regression parameter β of the factor Content was 37.08. This means that the average difference in ratings between violent and neutral stimuli was 74.16 points of the 100-point VAS. The unpleasantness ratings are depicted in Figure 2C.

Further, we found substantial evidence for the absence of a desensitizing VVG effect. Comparing a model where the fixed effect of Group*Content could be negative to a model where the effect was set to zero resulted in a BF of 0.151. Thus, participants of the violent game group did not show a decreased emotional response towards depictions of real and game violence. Moreover, testing the fixed effect of Group*Content*Context resulted in a BF of 0.094, indicating that there was also no desensitizing effect that was specific to depictions of game violence. When estimated without restrictions, the regression parameters associated with both interactions were positive, β = 2.28 for Group*Content, and β = 0.33 for Group*Content*Context. This means that, ostensibly, participants in the experimental group had a very weak tendency to rate violent images as more unpleasant than participants in the control group, contrary to expectations. However, note again that the Bayesian hypothesis test suggests that a model without these interactions provides a better explanation of the data. In summary, the behavioral data suggest that playing the violent video game did not emotionally desensitize participants towards violent images.”

– I suggest changing the title of the paper to remove the play on words and to better reflect the nature of the study if possible (short intervention, fMRI measures, etc.).

We have now changed the title to Neuroimaging and behavioral evidence that violent video games exert no negative effect on human empathy for pain and emotional reactivity to violence.

– Please specify whether violence from other characters was present for people playing the photograph version of the game.

All physical violence was removed from this version of the game. We have now emphasized this in the Methods section (lines 511-512):

“In the control game group, participants played a version of the game in which all violence was removed. The player character had no weapon, and could not hurt other characters in any way. They could also not be attacked by other characters, and there was no violence between non-player characters.”

– I have not found the "Data Availability Statement" required as per eLife policy and the data shared on the OSF (behavioural and ROIs extracted data) alone cannot be used to perform several analyses reported in the manuscript. The authors should consider maximally sharing the data analyzed in the manuscript and adequately documenting the data to favour reuse.

The data availability statement can be found in lines 663-670:

“Behavioral data, fMRI signal timecourses extracted from our regions of interest, task event timings, custom STAN code, and game images used in the emotional reactivity task are accessible at https://osf.io/yx423/. Unthresholded statistical maps are accessible at https://identifiers.org/neurovault.collection:13395. These include statistical maps from the analyses underlying the definition of our regions of interest, as well as the statistical maps from the frequentist analyses presented in Appendix 5. Full fMRI datasets from all participants are accessible at https://doi.org/10.5281/zenodo.10057633”

We have now also uploaded the full fMRI datasets from all participants to Zenodo: https://doi.org/10.5281/zenodo.10057633.

– The manuscript indicates that game images will be shared on request, but these seem to be available on the OSF repository of the study.

Thank you for spotting this mistake. We have corrected it.

– It would be helpful to state the sample size in the abstract and at the beginning of the Results section.

The sample size is stated in the abstract (lines 16-17: We recruited eighty-nine male participants without prior VVG experience). We have now also added it to the beginning of the Results section (lines 110-111):

“Forty-Five participants took part as part of the experimental group, and 44 participants as part of the control group.”

Reviewer #3 (Recommendations for the authors):The study has several strengths and raises important questions on the broader question of the psychological impact of violent video games. The use of a longitudinal design while controlling the amount of exposure to violent video games represents an advantage over past studies. It would be useful to know how the decision on the frequency and dosage of video games (2 weeks, 1 hour each day) was arrived at. It is not clear if this decision was governed by feasibility constraints alone, or if is there any prior literature that points to this dosage.

We thank you for your positive feedback. We have now also added an explanation regarding the quantity of exposure in the Methods section (lines 517-523):

“Due to the lack of other studies implementing a randomized experimental prospective design (except for Kühn et al. (2019), published while data collection was already on-going), there were no benchmarks for the amount and frequency of video game exposure for our study. We chose our regimen (seven one-hourly sessions over two weeks) as we considered this a substantial yet still feasible amount of exposure. We note though that the number of sessions, playing time per session, and total playing time were considerably higher than in previous studies reporting VVG effects on empathy (Arriaga et al., 2011; Carnagey et al., 2007; Engelhardt et al., 2011; Hasan et al., 2013).”

Second, the impact of playing violent (/non-violent) video games is measured using the empathy for pain paradigm in this study. It would be useful to note the test-retest reliability of the neural response in the empathy for pain paradigm.

We have now added estimates of the test-retest reliability of our measurements to the manuscript, in the new sections Results: Post-hoc analyses: Test-retest reliability (lines 229-241) and Methods: Post-hoc analyses: Test-retest reliability (lines 615-634). For our behavioral measurements, reliability was high to very high (⍴ ~ 0.7 for painfulness ratings, ⍴ ~ 0.9 for unpleasantness ratings). For neural responses, reliability was close to zero, though (⍴ ~ 0 for the three considered ROIs). To the best of our knowledge, our study was the first one to present the empathy for pain paradigm to the same sample of participants after a longer time frame. Thus, this surprising result provides valuable information on the limitations of this task respectively the neural measurements acquired in it, and certainly demands further research to investigate the factors influencing fMRI reliability.

We would like to emphasize, though, that a high test-retest reliability is not a precondition for the valid testing of group-level effects. For a group-level effect to be testable, it is only necessary that the mean of the dependent variable is consistently affected by the independent variable. It is not necessary that participants who show an above average level in the DV in one session also show an above average level in the second session, and vice versa. Otherwise, there would also be no point in independent-sample designs. Indeed, it has recently been discussed that highly robust cognitive tasks are bound to exhibit low test-retest reliability, as robust tasks are often characterized by low interindividual variation, and thus leave only little variance that can be explained by participant traits (Hedge et al., 2018). However, it must also be noted that low reliability does lead to lower power of repeated measures designs. As discussed in our previous response on power, the low reliability of the measured neural responses has resulted in suboptimal power of our tests on fMRI data.

We have added discussion of these points to the manuscript (lines 353-371):

“Lastly, and somewhat surprisingly, we found that the test-retest reliability of our neural covariates of empathy for pain were close to zero for all investigated ROIs. Knowing that an individual’s neural empathic response (BOLD activity for seeing somebody else in pain vs. in no pain) was above or below average in the first session provides little to no information about their relative response in the second session. To the best of our knowledge, our study was the first one to present the empathy for pain paradigm to the same sample of participants after a longer time frame. Thus, this surprising result provides valuable information on the limitations of this task respectively the neural measurements acquired in it, and certainly demands further research to investigate the factors influencing fMRI reliability (see also Elliott et al., 2020; Kragel et al., 2021). We would like to emphasize, though, that a high test-retest reliability is not a precondition for the valid testing of group-level effects. For a group-level effect to be testable, it is only necessary that the mean of the dependent variable is consistently affected by the independent variable. It is not necessary that participants who show an above average level in the DV in one session also show an above average level in the second session, and vice versa. Otherwise, there would also be no point in independent-sample designs. Indeed, it has recently been discussed that highly robust cognitive tasks are bound to exhibit low test-retest reliability, as robust tasks are often characterized by low interindividual variation, and thus leave only little variance that can be explained by participant traits (Hedge et al., 2018). However, it must also be noted that low reliability does lead to lower power of repeated measures designs, and that the low reliability of the measured neural responses has resulted in suboptimal power of our tests on fMRI data.”

Relatedly, is there any data to suggest that this neural response is sensitive to change in any external factors?

The empathy-for-pain paradigm that we used is well established, and the empathic neural response has been shown to be sensitive to be affected by various types of external factors. Among the factors that have been reported to modulate the response are ingroup-outgroup bias (Hein et al., 2010), perspective-taking (Lamm et al., 2007), placebo analgesia (Rütgen et al., 2015), and antidepressants (Rütgen et al., 2019). Especially the work using placebo analgesia, an intervention that traditionally has rather low effect sizes on first-hand pain, suggests that the lack of effects by the VVG is not due to the insufficient malleability of empathy and its neural bases, as operationalized and measured here. We clarified this point as follows in the discussion (lines 346-352):

“It may be argued that the empathy for pain paradigm and the associated behavioral and neural responses are so robust and resistant to changes by external factors that this may explain the lack of evidence for the effects of VVG play. This argument however would contradict a wealth of findings illustrating malleability of empathic responses using this and related designs, including with placebo analgesia (Rütgen et al., 2021; Rütgen, Seidel, Riečanský, et al., 2015; Rütgen, Seidel, Silani, et al., 2015), an intervention that usually shows low to moderate effect sizes as well (see e.g. Hein and Singer, 2008; Jauniaux et al., 2019; Claus Lamm et al., 2019; for review).”

1) It would be good to justify the choice of the smoothing kernel size.

We chose a kernel size of 4mm FWHM, as this is equal to twice the voxel size in every dimension, a common rule-of-thumb in fMRI research. This is now also explained in the manuscript (lines 559-560):

“The normalized functional images were smoothed with a Gaussian kernel of 4 mm full-width-at-half-maximum, which is equal to twice the voxel size on every axis.”

2) It is not entirely clear why the signal changes were extracted from ROIs only after thresholding the statistical maps. Arguably, if the hypotheses are about specific regions, and not about peak voxels that are significant at the level of the whole brain, then extracting the relevant metric from the entire ROI is likely to be more informative.

We now provide a better explanation of the motivation behind our masking procedure (lines 578-581):

“The aim of this masking procedure was to restrict analyses to those parts of the brain areas that are actually recruited by the task in question. We believe that this increased the sensitivity of our analyses, as we removed signals from voxels that are also part of these anatomical regions, but not actually recruited by the task.”

3) It is not clear whether the ROIs for the empathy-for-pain metric were based on the thresholded group maps from the first session only – or were they different between sessions?

We thank you for pointing out that this part was not entirely clear. It was shortly mentioned in lines 565-576 that the ROIs were only based on the data from the first session:

“To identify the regions in which empathic responses were reliably elicited independently of our experimental manipulation, we first analyzed the data from the first experimental session.”

To emphasize this point, we have now added the following (lines 583-584):

“Note that the ROIs, which were based on the signal from only the first session, were used to extract signals from both sessions.”

4) The use of two paradigms to measure the response to video games allows one to test the generalisability of the results. It would be useful to note if there is any relationship between the key metrics from these two paradigms, i.e. do individuals who show a greater empathy for pain (neurally/behaviourally) also show greater emotional reactivity?

Thank you for this interesting suggestion. We have now added analyses of cross-task correlations to the manuscript in the new sections Results: Post-hoc analyses: cross-task correlations (lines 266-272) and Results: Post-hoc analyses: cross-task correlations (lines 654-662). They are also reproduced at the end of this response letter. In short, we found small-to-medium correlations (.227 –.280) between the behavioral measures of the empathy for pain task and the emotional reactivity task. However, we could not observe any noteworthy correlations between the neural measurements. This might reflect that the two tasks tap into two distinct, yet related processes: while one measures empathy for pain, the other measures responses to visually upsetting situations.